# Improving fire severity prediction in south-eastern Australia using vegetation specific information

Kang He[1,2], Xinyi Shen[3], Cory Merow[2,4], Efthymios Nikolopoulos[5], Rachael V. Gallagher[6], Feifei Yang[1,2], Emmanouil N. Anagnostou[1,2]

[1]Department of Civil and Environmental Engineering, University of Connecticut, Storrs, CT 06269, USA

[2] Eversource Energy Center, University of Connecticut, Storrs, CT 06269, USA

[3]School of Freshwater Sciences, University of Wisconsin, Milwaukee, Milwaukee, WI, 53204, USA

[4]Department of Ecology and Evolutionary Biology, University of Connecticut, Storrs, CT 06269, USA

[5]Department of Civil and Environmental Engineering, Rutgers University, Piscataway, NJ 08854, USA

[6]Department of Biological Sciences, Macquarie University, North Ryde, NSW 2109, Australia

*Correspondence to*: Emmanouil N. Anagnostou (emmanouil.anagnostou@uconn.edu)

**Abstract.** Wildfire is a critical ecological disturbance in terrestrial ecosystems. Australia, in particular, has experienced increasingly large and severe wildfires over the past two decades while globally fire risk is expected to increase significantly due to the projected increase in extreme weather and drought condition. Therefore, understanding and predicting fire severity is critical for evaluating current and future impacts of wildfires on ecosystems. Here, we firstly introduce a vegetation-type specific fire severity classification applied on satellite imagery, which is further used to predict fire severity using antecedent drought conditions, fire weather (i.e., wind speed, air temperature and atmospheric humidity), and topography of the fire season (November to March). Compared with fire severity maps from Fire Extent and Severity Mapping (FESM) dataset, we find fire severity prediction results using the vegetation-type specific thresholds show good performance in extreme and high severity classification with accuracy of 0.64 and 0.76, respectively. Based on a 'leave-one-out' cross-validation experiment, we demonstrate high accuracy for both the fire severity classification and the regression using a suite of performance metrics: determination coefficient ($R^2$), mean absolute error (MPE) and root mean square error (RMSE), which are 0.89, 0.05, and 0.07, respectively. Our results also show that the fire severity prediction results using the vegetation-type specific thresholds could better capture the spatial patterns of fire severity, and has the potential to be applicable for seasonal fire severity forecast due to the availability of seasonal forecasts of the predictor variables.

Keywords: Fire severity; Normalized Burning Ratio; Random Forest; Vegetation type; Severity classification.

## 1 Introduction

Fire is recognized as a critical disturbance in ecosystems, which shapes vegetation across several continents (Archibald et al., 2013; Gill, 1975; Giglio et al., 2010; Gomez et al., 2015). In recent decades, wildfires have affected extensive areas in forests and woodlands across the globe, including those in Australia where over 10 million hectares were burned in the 2019-2020 fire season (from November to March, Gallagher et al. 2021). These fires are considered unprecedented in contemporary Australian fire history (Nolan et al., 2020; Shine, 2020), and more severe fires are expected in the future due to the impacts of

climate change on fire-weather and dynamics (Hennessy et al., 2005). Changes in fire conditions are also anticipated globally (Abatzoglou et al. 2019). Therefore, predicting fire characteristics – such as severity – will be essential for evaluating current and future impact of wildfires on ecosystems worldwide.

Fire severity, defined here as the magnitude of change in vegetation associated with fire, is routinely used to describe the impact of wildfires on vegetation, soil and wildlife (Lentile et al. 2006; Keeley 2009). Field survey and remote sensing-based evaluations of burn severity are commonly used by fire scientists and managers. Field survey-based evaluations involve assessing the amount of biomass consumed (Keeley, 2009), measuring the changes in vegetation height (Wang and Glenn, 2009) or surface fuel consumption (Boby et al., 2010; Hudak et al., 2013). By contrast, remotely sensed evaluations of burn severity use satellite imagery to quantify the magnitude of vegetation changes between pre-fire and post-fire conditions, in terms of the changes in surface reflectance (Holden et al., 2009; Miller et al., 2009; Soverel et al., 2010) (e.g. the difference between pre- and post-fire Normalized Burn Ratio (dNBR), Keeley, 2009).

Statistical approaches, which incorporate factors such as topography, weather and water availability provide insight into possible drivers of fire severity (Morgan et al., 2014). For instance, Bradstock et al. (2010) investigated the effects of weather, fuel and terrain on fire severity in south-eastern Australia. They found weather was the predominant influence on fire severity while the influence of terrain was stronger under moderate conditions. Similarly, a study by Collins et al. (2013) examined the relationships between environmental variables (i.e., fire weather, topography and fuel age) and fire severity in south-eastern Australia and whether it can be modified by increasing mean annual precipitation. They concluded that the relationships between crown fire and weather, topography and fuel age were largely unaltered across the precipitation gradient. Collins et al. (2019) also examined the relative effect of fire weather, drought severity and landscape features (i.e., topography, fuel age, vegetation type) on the occurrence of fire refugia in south-eastern Australia. They found that the fire weather and drought severity were the primary drivers of the occurrence of fire refugia, moderating the effect of landscape attributes. Furthermore, Clarke et al. (2014) investigated fire severity control factors, including landscape/vegetation or weather, providing evidence that even though strong weather controls, fire history, terrain and vegetation shape the immediate effect. In addition, Bowman et al. (2021) demonstrated that overwhelming dominance of fire weather in driving complete scorch or consumption of forest canopies in natural and plantation forests in the 2019-20 megafires.

Despite the emerging evidence that statistical modelling with multiple biophysical and environmental predictor variables can provide high accuracy estimates of fire severity, this technique is not widely adopted in major areas of known fire risk. One such region is the southeast coast of Australia which is subject to annual fire seasons (from November to March, Collins et al., 2022) vary in extent and severity and has a high richness of endemic plant species adapted to particular fire regimes (Gallagher et al., 2021). Besides, an accurate representation of fire severity levels is important for managing and mitigating the effects of wildfires, both in terms of emergency response and long-term ecological recovery. Existing fire severity classification schemes rely on the in-situ measurements of Composite Burn Index (CBI, Key and Benson, 2006; Lutes et al., 2006) and/or aerial photographs identification (Collins et al., 2018; Dixon et al., 2022) which are available for certain regions and for limited vegetation types under certain climate (Eidenshink et al., 2007; Keeley et al., 2009; Tran et al., 2018). However, obtaining

CBI and interpreting aerial photographs are labor-intensive and time-consuming, especially over large areas, while inferring fire severity levels directly from satellite-derived dNBR can be more efficient for large-scale applications, yet no dNBR-based fire severity classification scheme exists for regions such as the southeast coast of Australia, which is subject to recurring annual wildfires and varies greatly in vegetation types with high richness of endemic plant species adapted to particular fire regimes (Gallagher et al., 2021).

Understanding current and predicting future fire severity in eastern Australia is critical for evaluating the potential for increased extinction risk due to recurrent high severity fires (Enright et al. 2015) and is important for supporting ecologically informed fire management (Clarke et al. 2019). Therefore, the predictor variables involved in the fire severity model should be accessible for both historical events and projected future events (e.g. seasonal, climate).

In this study, we newly propose a vegetation specific fire severity classification scheme for predicting fire severity and demonstrate its performance across the Australian state of New South Wales (NSW). Using drought conditions, vegetation type, and fire weather conditions during the fire season as input, our modelling approach applies the Random Forest (RF) classification method to predict the dNBR – an indicator of burn severity derived from Landsat imagery. We demonstrate model performance based on 20 years of wildfire data from NSW through a leave-one-year-out cross-validation experiment.

## 2 Study area

New South Wales (NSW) in south-eastern Australia (Figure 1) occupies a subtropical-temperate climate region with relatively mild weather and distinctive seasons (e.g., hot summers and cold winters) (Speer et al., 2009). Mean annual and extreme temperatures are highest in the northwest of the state whereas average maximum temperatures in coastal areas range from 26 °C to 16 °C, while the average minimum temperature falls between 19 ° and 7 °C. There is a strong precipitation gradient from east to west across the state, with annual precipitation on the eastern coast ranging between 600 mm/year and 1200 mm/year decreasing to generally less than 180 mm/year in the north west of the state Vegetation across the study region is predominantly wet and dry sclerophyll forests, although is interspersed with areas of rainforest, woodlands and coastal heath (Keith 2004).

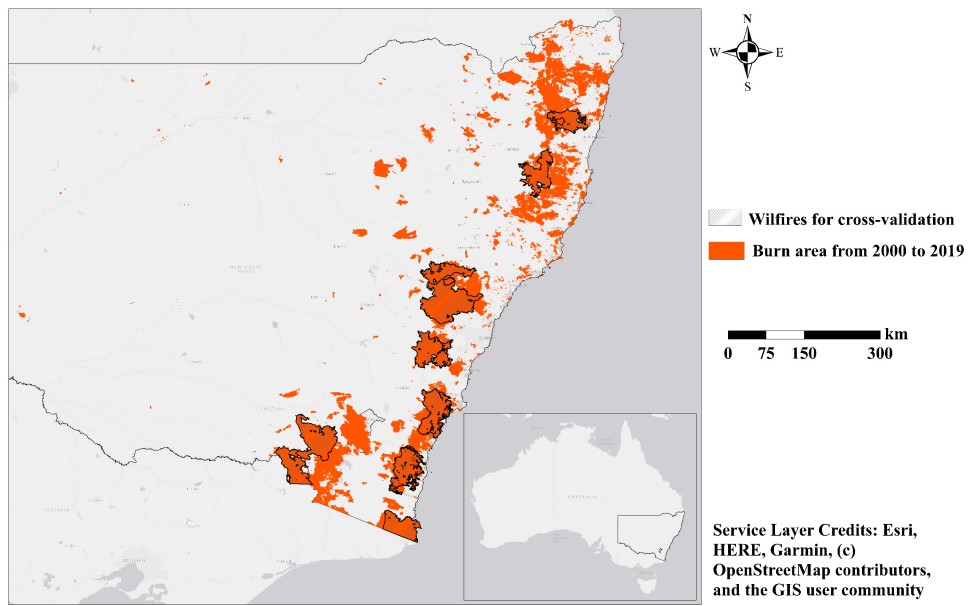

Figure 1. Locations of study wildfires over New South Wales (NSW), Australia. The burn area is from NSW National Parks and Wildlife Service (NPWS) Fire History – Wildfire and Prescribed Burns dataset.

**3 Data and method**

**3.1 Model Input and output**

**3.1.1 Fire extent**

The spatial extent of annual fires between 2000 to 2019 is accessed from the NSW National Parks and Wildlife Service (NPWS) Fire History – Wildfire and Prescribed Burns dataset (https://datasets.seed.nsw.gov.au/dataset/fire-history-wildfires-and-prescribed-burns-1e8b6), produced by the Department of Planning, Industry and Environment. The NPWS Fire History is a spatial polygon layer, with each polygon recording the boundary, start date, end date, and burn area. We use the NPWS polygons whose burn areas are greater than 1 km$^2$ as the mask to include only the fire impacted areas. While this dataset is unlikely to be a complete record of all fire events, it represents the largest single repository of fire extent data in NSW.

**3.1.2 Fire severity**

As a widely used fire severity index, the dNBR is calculated by subtracting the post-fire NBR raster from the pre-fire NBR raster as in Eq (1) (Keeley, 2009):

$$dNBR = PrefireNBR - PostfireNBR \qquad (1)$$

The formula of NBR is similar to the normalized difference vegetation index (NDVI), except that it uses near-infrared (NIR) and shortwave-infrared (SWIR) bands, as written in Eq (2) (García and Caselles, 1991; Key and Benson, 2006). NBR can be

computed by the Thematic Mapper (TM) and Enhanced Thematic Mapper Plus (ETM+) sensors on using Band 7 as the short-wave infrared (SWIR) and Band 4 for Landsat 4-7 and Band 5 for Landsat 8 as the near infrared (NIR) reflectance, respectively. While in Sentinel-2, SWIR and NIR are represented by Band 8 and Band 12, respectively.

$$NBR = \frac{NIR - SWIR}{NIR + SWIR} \qquad (2)$$

We calculate the dNBR within the fire boundaries from Landsat and Sentinel archive imagery, using the start date and end date to determine the pre-fire and post-fire dates. In this study, the pre-fire NBR (preNBR), is used as a proxy of the initial condition of vegetation. The calculation of a dNBR-image is described as follows: (1) determine a individual fire from NPWS Fire History; (2) collect the most recent Landsat images based on the tags demarcating the start and end times of each individual fire; (3) apply a cloud- and snow-masking algorithm to remove snow, clouds, and their shadows from all imagery based on each sensor's pixel quality assessment band; (4) use the auxiliary satellite images (e.g., Sentinel-2) to fill the blank pixels in the cloud-free images from step (3) to obtain the pre and post NBR composites; (5) subtract pre- and post-NBR images to create a dNBR composite with the smallest possible cloud and shadow extent. The dNBR typically ranges from -2 to +2, with high positive values indicating severe burn damage where the vegetation has been completely consumed. Values around zero suggest either unburned areas or areas where the fire had a very low impact. Negative values can indicate an increase in vegetation, which might be due to vegetation recovery over time or errors in the analysis.

### 3.1.3 Vegetation

Vegetation composition and structure are expected to influence fire propagation and severity (Collins et al., 2007) and the vegetation type is also used as a proxy for vegetation structure (Hammill et al., 2006). The dominant vegetation over NSW is wet and dry sclerophyll forests (Keith 2004). Wet sclerophyll forests can be divided into two subgroups (the shrubby sub-formation and the grassy sub-formation), which have a tall canopies dominated by Eucalyptus and a monophyllous understory (https://www.environment.nsw.gov.au/threatenedSpeciesApp/VegFormation.aspx?formationName=Wet+sclerophyll+forests+(grassy+sub-formation) ). Two sub-formations of dry sclerophyll forests also occur: shrub/grass and shrubby. This study focuses on burn severity for the dominant sclerophyll forests (Figure 1). The vegetation map is intersected with NPWS polygons to identify the areas where sclerophyll forests have previously burned.

### 3.1.3 Topography

Prior studies report strong control of topography on burn severity, by influencing fire behavior, fuel moisture, and water balances (Fang et al., 2018, Harris and Taylor, 2015, Holden et al., 2009). Therefore, we include three topographic measures from Shuttle Radar Topography (SRTM, https://www2.jpl.nasa.gov/srtm/world.htm ), elevation (DEM), slope (Slope), and Topographic Position Index (TPI). TPI helps in identifying landform features such as ridges, valleys, slopes, and plateaus (Weiss, 2001). Positive TPI values indicate locations that are higher than the average of their surroundings (e.g., hilltops or

ridges), while negative TPI values indicate locations that are lower than their surroundings (e.g., valleys or depressions). Values
close to zero may represent flat areas or slopes.

### 3.1.3 Weather

In addition to fuels and terrain, weather is another important factor in wildfires. The McArthur Forest Fire Danger Index (FFDI,
McArthur 1967) is an empirical relationship comprising the short-term meteorological conditions and the long-term drought
factor (Dowdy et al. 2009). The FFDI is currently used operationally by the Australian Bureau of Meteorology (BoM) to
produce fire weather warnings to authorities, which is defined as:

$$FFDI = 2 \times e^{(-0.45 + 0.897 lnDF - 0.0345 RH + 0.038T + 0.0234V)} \tag{3}$$

where DF is the drought factor; and    RH, T and V represent the relative humidity, surface air temperature and wind velocity,
respectively. In this study, we extract daily temperature, relative humidity and wind speed data from the ERA5-Land global
dataset over the burn areas (https://cds.climate.copernicus.eu/cdsapp#!/dataset/reanalysis-era5-land?tab=form ).
The DF is estimated using the Keetch–Byram Drought Index (KBDI, Keetch and Byram 1968). KBDI is a continuous reference
scale describing the dryness of the soil and duff layers. The index increases for each day without rain and decreases when it
rains. KBDI is world widely used for drought monitoring for national weather forecast, wildfire prevention. KBDI over burnt
areas can be accessed in Takeuchi et al. (2015). The daily FFDI and KBDI values for the day prior to the start of the wildfires
are used as the predictors in predicting burn severity, owing to the strong correlation in time between extreme values of the
FFDI and  the start of the wildfires [Dowdy et al., 2009]Using the most potential extreme FFDI, indicating the extreme weather
conditions, in the period leading up to a wildfire could address the impact of weather on wildfire risk.

### 3.2 Method

We newly propose an alternative way to determine the optimal thresholds in fire severity classification for different vegetation
types. The dNBR of all burnt pixels for each vegetation type are collected and a set of dNBR values at the quantiles varying
from 5% to 35% representing the threshold for low severity classification, quantiles varying from 35% to 65% representing
the threshold for moderate severity classification, and quantiles varying from 65% to 95% representing the threshold for high
severity classification. For example, a classified burn severity sample can be obtained using the thresholds for high, moderate,
and low severity at 85% quantile, 55% quantile and 25% quantile, respectively. Secondly, a fire severity prediction model is
developed for each severity category based on the fire severity classification results, to provide the numeric prediction of
dNBR.

### 3.2.1 Fire severity classification by RF

Random Forest is developed as an extension of the classification and regression tree (CART) to improve the accuracy and
stability of the CART model (Breiman 2001). The steps of the RF algorithm are briefly summarized as: (i) randomly generate
*ntree* bootstrap samples of the original data. The elements not selected are referred to as 'out of bag' (OOB) samples. (ii) for
each split, randomly select *m_try* predictors of the original predictors and choose the best predictor among the *m_try* predictors
to partition the data. (iii) predict new data (OOB elements) by averaging predictions of the *ntree* trees; and (iv) the OOB
samples are used to estimate the prediction error. The RF can also provide a measurement of variable importance. One of the
approaches is to look at the increase in the OOB estimate error when the specific predictor variable is randomly permuted and
other predictors are constant. The more the error increases, the more important the variable is. These variable importance
values are used to rank the predictors in terms of their relative contribution to the model. The RF model was generated using
the package randomforest in R (https://cran.r-project.org/web/packages/randomForest/ ).

### 3.2.2 Fire severity prediction by XGboost

For the regression model, we implement the Extreme Gradient Boosting (XGBoost) algorithm, one of the most popular
supervised machine learning algorithms proposed by Chen et al. (2015). XGBoost employs a gradient boosting framework
that iteratively trains a sequence of weak prediction models and combines them into a strong model. In addition to gradient
boosting, XGBoost implements several advanced features, including regularization techniques to prevent overfitting, parallel
processing to speed up training, and built-in support for missing data (Chen and Guestrin, 2016 ). In the XGBoost algorithm,
complex interactions are modeled, and other complexities such as missing values in the predictors are managed without almost
any loss of information. Selection of features is performed by a combination of parameters (e.g., number of iterations, learning
rate) and the unique combinations of each attribute in the training data set. The XGBoost model is generated using the package
xgboost in R (https://cran.r-project.org/web/packages/xgboost/ ).

### 3.2.3 Calibration and validation

The fire severity classification maps from Fire Extent and Severity Mapping (FESM,
https://datasets.seed.nsw.gov.au/dataset/fire-extent-and-severity-mapping-fesm) in period from 2016 to 2019 are used as the
independent source to validate the fire severity classification results based on the proposed method. To evaluate the model's
performance, we also use "leave -one group-out" for training and validation. The fire samples from 2000 to 2019 are firstly
divided into 20 subsets depending on the year the fire occurred, and this holdout method is repeated 20 times. Each subset
represents the samples from the wildfire with the largest burn area in the corresponding year. Secondly, at each time, one of
the 20 subsets is used as the testing set, and the remaining 19 subsets are put together to form the training set. Thirdly, the
average error across all 20 trials is computed. The advantage of this cross-validation method is that it gives us an indication of
how well the model would do when making new predictions for data it has not already seen.
For performance evaluation of multiclass event classification, accuracy is expressed as the proportion of correctly predicted
events over all predicted events, which is calculated as Eq (4):

$$Accuracy = \frac{Number\ of\ correct\ predictions}{Number\ of\ all\ predictions} \tag{4}$$

While precision is expressed as the proportion of events correctly predicted as label X (low, moderate, or high) over all events predicted as label X (Eq (5)).

$$Precision = \frac{True\ Positive}{True\ Positive + False\ Positive} \tag{5}$$

in which True Positive represents the situation both observation and prediction are labelled as X, False Positive represents observation is not labelled as X but prediction as label X.

Recall is calculated as:

$$Recall = \frac{True\ Positive}{True\ Positive + False\ Negative} \tag{6}$$

in which False Negative represents the situation observation is label X but prediction is not label X.

Combining metrics of Precision and Recall, the F1 Score is the harmonic mean of Precision and Recall. The F1 Score gives equal weight to Precision and Recall. A maximized F1 Score could create a balanced classification model, and is calculated as follows:

$$F1\ score = 2 * \frac{Precision * Recall}{Precision + Recall} \tag{7}$$

The coefficient of determination ($R^2$) is used to measure how well the prediction agreed with the actual values. The formula of $R^2$ is described as:

$$R^2 = \frac{1}{n}\sum_{i=1}^{n} \frac{(o_i - \frac{\sum_{i=1}^{n} o_i}{n})(p_i - \frac{\sum_{i=1}^{n} p_i}{n})^2}{o_i p_i} \tag{8}$$

Where $o_i$ and $p_i$ represent the actual and predicted values for sample i; n is the total number of samples. The higher $R^2$ indicates better fit of the model predictions to the actual values with best value of 1.

The mean absolute error (MAE) the mean relative error, the lower MAE is, the better the model performed.

$$MAE = \frac{\sum_{i=1}^{n}|p_i - o_i|}{n} \tag{9}$$

The root mean square error (RMSE) is used to quantify the random component of the error. The lower RMSE indicates better model performance.

$$RMSE = \sqrt{\frac{\sum_{i=1}^{n}(p_i - o_i)^2}{n}} \tag{10}$$

**4 Results**
**4.1 Fire severity of burnt vegetation**
Over the past 20 years, wildfire history databases managed by government agencies indicate that approximately 112,590 km$^2$
have been recorded as affected by fires in NSW, of which, almost 53,830 km$^2$ burned during the 2019-20 megafires (Figure
2). This dataset indicates that the annual burn area is typically below 5,000 km$^2$, but in exceptional years such as 2002 and
2003, the affected area can reach more than 10,000 km$^2$. The affected area from the 2019-20 fires is approximately 10 times
larger than those in other years from 2004 to 2018.

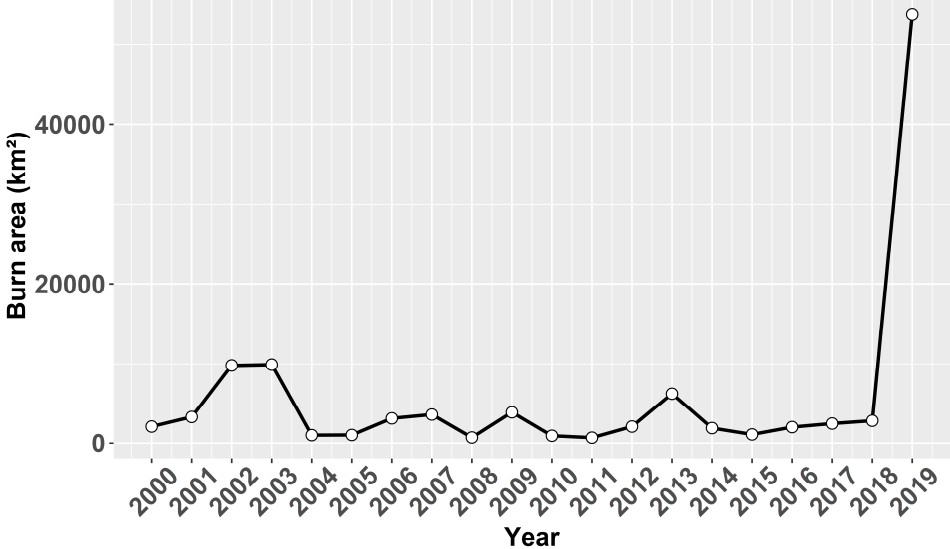

Figure 2. Annual burnt area (km$^2$) across New South Wales, in south-eastern Australia.

Among the burnt area, the fractions of vegetation types are shown in Figure 3 (a). The dry sclerophyll forests (shrubby
subformation) accounted for the largest proportion of the burnt area (32.1%), followed by the dry sclerophyll forests
(shrub/grass subformation) which account for 16%. The wet sclerophyll forests (grassy subformation) occupy 14.2% of the
burnt area, while for the wet sclerophyll forests (shrubby subformation) the proportion is 11%. Specifically, the cleared area
accounted for 11.3% of the burnt area, approximately equal to those of the wet sclerophyll forests (shrubby subformation).
Other vegetation types largely affected by the wildfires are grassy woodlands, rainforest and heathlands, the proportion of
which are 6.7%, 2.5% and 2%, respectively. The distribution of fire severity indicated by dNBR for each vegetation type is
displayed as Figure 3 (b). These boxplots in Figure 3 (b) show that the fire severity varies significantly with vegetation type,
demonstrating that the vegetation specific thresholds should be applied in fire severity classification. For example, the fire
severity of cleared areas is overall the smallest while the fire severity of heathland shows the overall largest. The fire severity

 varies even for the major vegetation type with different subgroups, for instance, the fire severity of dry sclerophyll forests with

shrubby subformation is larger than the fire severity of dry sclerophyll forests with shrub/grass subformation.

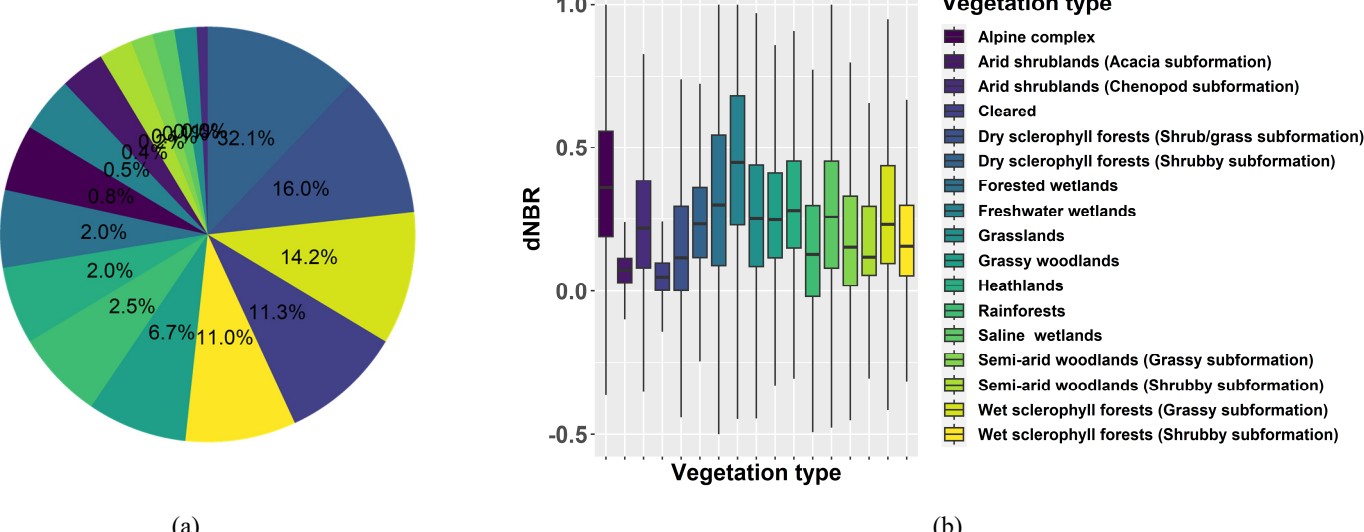

(a)           (b)

Figure 3. (a) The proportion of burnt area and (b) the distribution of fire severity grouped by vegetation type, over NSW from 2000 to 2019


**4.2 Threshold determination for fire severity classification**

Given the variability shown in Figure 3 (b), we proposed an alternative way to determine the optimal thresholds in fire severity
classification for different vegetation types. To determine these thresholds the dNBR of all burnt pixels for the vegetation type
were collected and a set of dNBR values at the quantiles from 0.05 to 0.95 are used as the candidates of thresholds for the fire
severity classification. The classified samples using the threshold of dNBR at the quantiles are imported as the training set in
RF models and the OOB estimate of error rate is recorded for the training samples. Figure 4 (a), (b), (c) and (d) show the
variations of OOB estimate of error rate changes with thresholds of dNBR at the quantiles varying from 5% to 35% (low
severity threshold)/35% to 65% (moderate severity threshold), when the high severity threshold are set as the dNBR values at
the 65%, 75%, 85% and 95% quantiles, respectively. The optimal thresholds are determined when the lowest OOB estimate
of error rate is found. For example, for dry sclerophyll forests (shrubby subformation), the thresholds for high, moderate and
low severity classification are 0.55 (85% quantile), 0.38 (55%) and 0.20 (25%), respectively. It is important to be aware that
the classification step is merely used to improve the consecutive regression accuracy, rather than the final severity
categorization result. The choice of threshold in this step therefore will not affect severity categorization. The categorization
will be solely based on predicted severity value, using user defined thresholds.

**OOB estimate of error rate at 0.65 quantile for high severity classification**

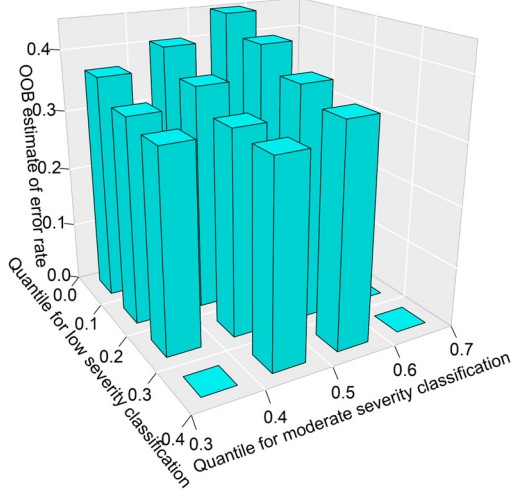

**OOB estimate of error rate at 0.75 quantile for high severity classification**

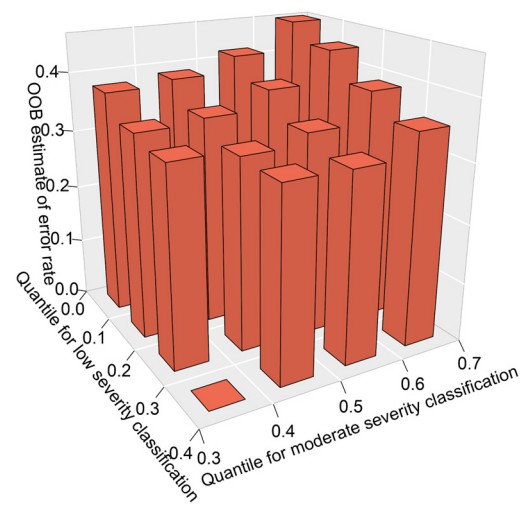

**OOB estimate of error rate at 0.85 quantile for high severity classification**

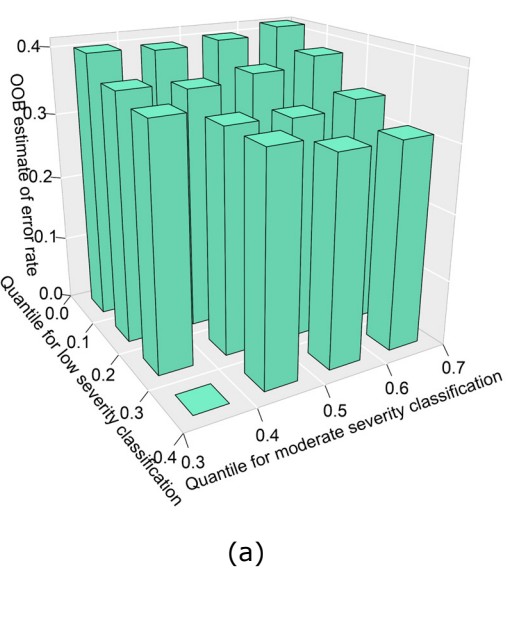

**OOB estimate of error rate at 0.95 quantile for high severity classification**

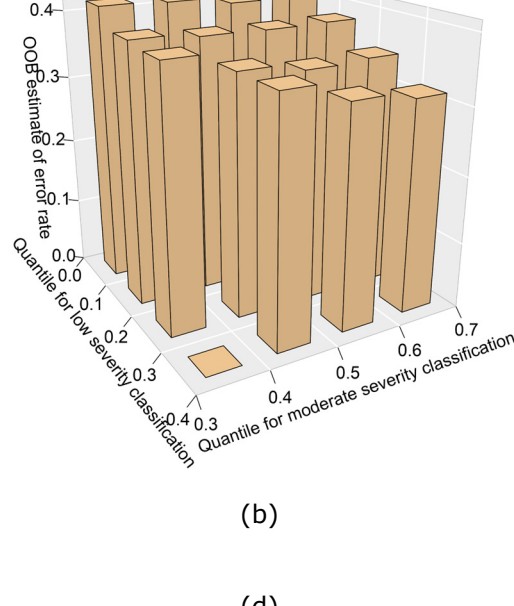

(a)                                                   (b)

(c)                                                   (d)

Figure 4. Variations of OOB estimate of error rate changes with thresholds of dNBR at the quantiles varying from 5% to 35% (low severity threshold)/35% to 65% (moderate severity threshold), when the high severity threshold are set as the dNBR values at the (a) 65%, (b) 75%, (c) 85% and (d) 95% quantiles.


The thresholds of dNBR for fire severity classification for different vegetation types are determined by the proposed method
and the results are presented in Table 1. It is shown that the thresholds vary significantly with vegetation type. For example,
for rainforests when dNBR of burnt area is around 0.20, this area should be classified as high severity. However, the burnt
area with the same dNBR (0.20) would be classified as moderate severity when wildfire burns over other vegetation types.
This difference is also found in the major vegetation type within different subgroups. A burn area with dNBR around 0.53 is
classified as extreme high severity when fire burns over wet sclerophyll forests (grassy subformation), while this burn area is
classified as high severity when fire burns over wet sclerophyll forests (shrubby subformation). The differences in
classification thresholds are more significant between dry sclerophyll forests with shrub/grass subformation and shrubby
subformation. The thresholds for high severity classification are 0.44 and 0.55 for burnt area over dry sclerophyll forests
(shrub/grass subformation) and dry sclerophyll forests (shrubby subformation), respectively. These results indicate that using
the vegetation specific thresholds would obtain more reasonable fire severity classification results, while a lot of
misclassifications are found when applying fixed thresholds in fire severity classification without considering the variations in
vegetation cover.
Table 1. Thresholds of dNBR for fire severity classification by vegetation type.

| Vegetation | Low | Moderate | High | Extreme |
|---|---|---|---|---|
| Rainforests | < 0.05 (25%) | 0.05 - 0.18 (25%-45%) | 0.18 – 0.41 (45%-75%) | > 0.41 (75%) |
| Wet sclerophyll forests (Shrubby subformation) | < 0.15 (35%) | 0.15 - 0.34 (35%-55%) | 0.34 - 0.56 (55%-85%) | > 0.56 (85%) |
| Wet sclerophyll forests (Grassy subformation) | < 0.17 (35%) | 0.17 - 0.34 (35%-55%) | 0.34 - 0.52 (55%-85%) | > 0.52 (85%) |
| Grassy woodlands | < 0.15 (35%) | 0.15 - 0.36 (35%-55%) | 0.36 - 0.55 (55%-85%) | > 0.55 (85%) |
| Dry sclerophyll forests (Shrub/grass subformation) | < 0.12 (15%) | 0.12 - 0.26 (15%-45%) | 0.26 - 0.44 (45%-75%) | > 0.44 (75%) |
| Dry sclerophyll forests (Shrubby subformation) | < 0.20 (25%) | 0.20 – 0.38 (25%-55%) | 0.38 – 0.55 (55%-85%) | > 0.55 (85%) |
| Heathlands | < 0.26 (35%) | 0.26 – 0.40 (35%-55%) | 0.40 – 0.57 (55%-75%) | > 0.57 (75%) |


**4.3 Fire severity prediction results**
The performance of vegetation specific thresholds and the importance of vegetation type are validated by the cross-validation
in the RF model. Figure 5 (a) and (b) show the relative importance of variables in the RF based on samples classified by
vegetation specific thresholds and fixed thresholds, respectively. The error bar represents the standard deviation (sd) of relative
importance in RF models in the cross-validation experiments. The preNBR is the most influential variable with relative
importance around 28% and sd around 7%. The FFDI also plays an important role in the model with relative importance and
sd of 21% and 6%, respectively. The KBDI shows close relative importance to those of FFDI, the values of mean relative
importance and sd are 19% and 5% respectively. While for vegetation type, the relative importance (13%) is higher than those
of topographic variables when the vegetation specific thresholds are applied. The sd of vegetation type is the largest (9%),
owing to the differences in vegetation diversity in the training samples.

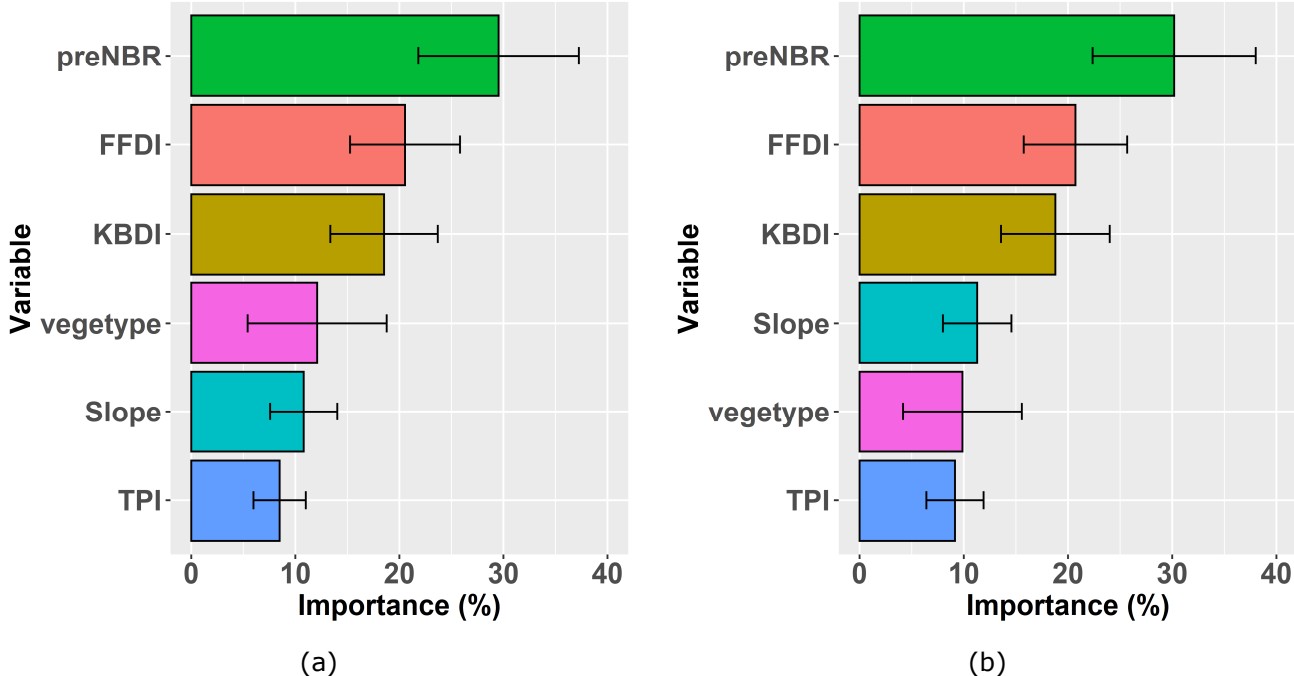

(a)                                                          (b)

Figure 5. Relative importance of variables in RF models based on samples classified by (a) vegetation specific thresholds and (b) fixed thresholds.


The confusion matrix of the fire severity classification results is shown in Table 2. More samples are classified into extreme
high severity classification when applying vegetation specific thresholds than those using fixed thresholds. Similarly, more
samples are classified into low severity while implementing fixed thresholds than vegetation specific thresholds. This indicates
that using fixed thresholds without considering the vegetation type tends to underestimate the fire severity levels. While for
the performance of fire severity prediction, most events of extreme high severity are correctly identified by the RF model
trained by samples classified by vegetation specific thresholds while more misclassified extreme high severity and high
severity events are predicted by the RF model trained by samples classified by fixed thresholds.

Table 2. Confusion matrix of prediction results based on RF model trained by samples classified by vegetation specific and
fixed thresholds.

| Vegetation specific | | | | Fixed | | | | |
|---|---|---|---|---|---|---|---|---|
| | Extreme | High | Moderate | Low | | Extreme | High | Moderate | Low |
| Extreme | 52680 | 22782 | 813 | 9 | Extreme | 36573 | 24573 | 1755 | 30 |
| High | 4749 | 94899 | 17265 | 171 | High | 3930 | 64740 | 21498 | 471 |

| | | | | | | | | |
|---|---|---|---|---|---|---|---|---|
| Moderate | 501 | 20487 | 103536 | 3948 | Moderate | 852 | 19794 | 94857 | 8739 |
| Low | 147 | 1422 | 22239 | 36897 | Low | 357 | 2754 | 31299 | 70347 |


The overall classification accuracy calculated by equation (4) is 0.75 and 0.69, for RF models trained by samples classified by
vegetation specific and fixed thresholds, respectively. Figure 6 (a), (b) and (c) show the Precision, Recall and F1 score of event
severity classification results for each class label calculated by equations (5) – (7). The Accuracy, Precision, Recall results and
F1 Score close to 1 indicate accurate classification results. For the classification metrics of each class label, the high severity
events class exhibit the best Precision (0.85) relative to the moderate (0.76) and extreme high severity event classes (0.68),
while the Recall and F1 score for high severity events class are 0.64 and 0.73, respectively. The extreme high severity events
class exhibit the best Recall (0.89) relative to the other two classes, and the Precision and F1 score are 0.68 and 0.77,
respectively. The performances of fire severity classification are worse for the RF model trained by samples classified by the
fixed thresholds, with lower precision, recall and F1 score.

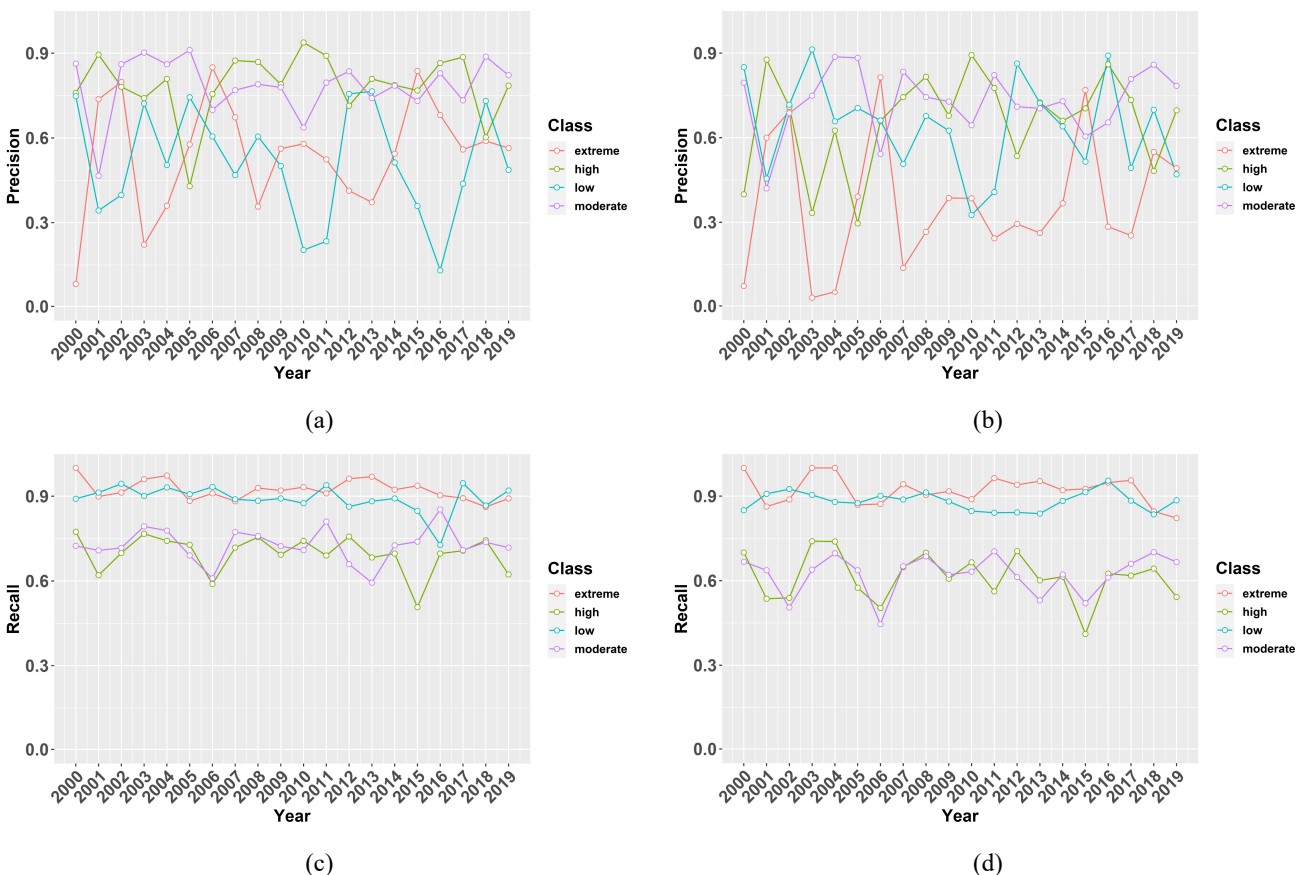

(a)

(b)

(c)

(d)

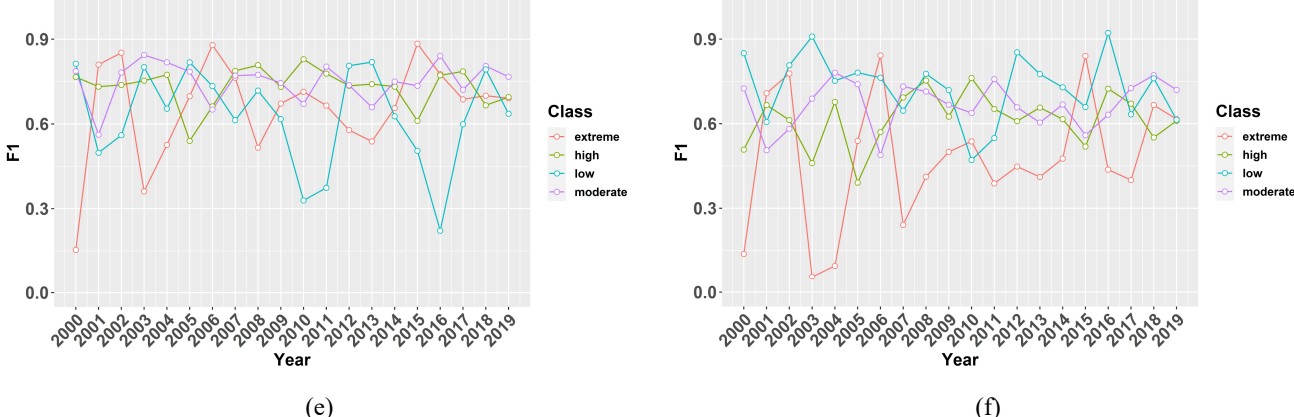

(e)                                                                    (f)

Figure 6. Results of Precision for predictions based on (a) vegetation specific thresholds and (b) fixed thresholds; The results of Recall for predictions based on (c) vegetation specific thresholds and (d) fixed thresholds; The results of F1 score for predictions based on (e) vegetation specific thresholds and (f) fixed thresholds;



Figure 7 displays the fire severity maps for the 2016, 2017, 2018 and 2019 wildfires in NSW from FESM, along with fire
severity predictions based on vegetation specific and fixed thresholds. For the wildfire in 2016, predictions based on vegetation
specific thresholds show similar spatial patterns of fire severity to those from FESM, while predictions based on fixed
thresholds significantly underestimate the fire severity in the high and extreme fire severity areas of the FSEM. Similarly for
the wildfire in 2018, predictions based on fixed thresholds significantly underestimate high and extreme severity compared to
the FESM map, while predictions based on vegetation specific thresholds slightly underestimate extreme severity. For the
wildfire in 2017, both the FESM and predictions display similar spatial distributions of fire severity level with predictions
based on fixed thresholds presenting more low severity compared to FESM map. For the wildfire in 2019, however, predictions
based on fixed thresholds tend to overestimate the fire severity as extreme in regions found to be high severity in FESM map,
while predictions based on vegetation specific thresholds agreed better with FESM map.

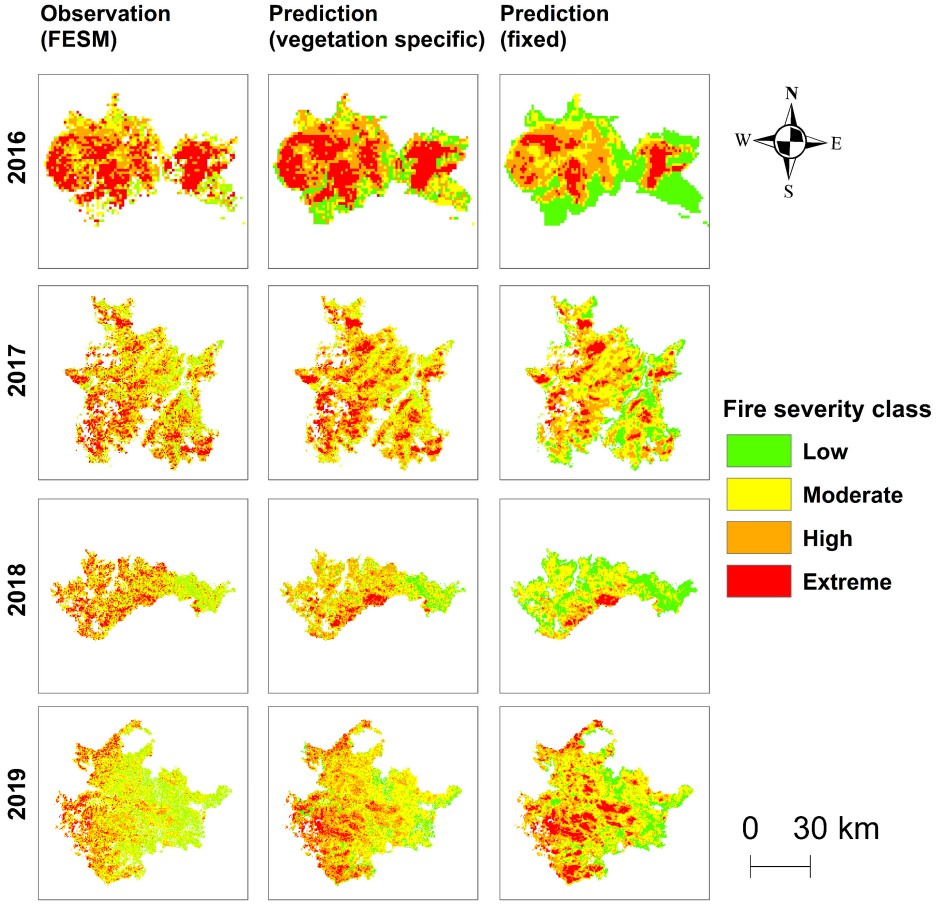

Figure 7. Fire severity classification maps from FESM and predictions based on vegetation specific and fixed thresholds for wildfires in 2016 to 2019 in NSW.

Table 3 shows the confusion matrix for fire severity classification between FESM and predictions based on vegetation specific and fixed thresholds. It is noted that predictions based on vegetation specific thresholds exhibit better ability of classing extreme and high severity with accuracy of 0.64 and 0.76, respectively. While the classification accuracy for extreme and high severity of predictions based on fixed thresholds are 0.21 and 0.39, respectively. Predictions based on vegetation specific thresholds also have better accuracy of classifying moderate severity with value of 0.62, compared to those based on fixed thresholds with value of 0.47. Both predictions based on vegetation specific and fixed thresholds show poor performance in classifying low severity, with accuracy of 0.24 and 0.26 respectively. The overall classification accuracy for predictions based on vegetation specific thresholds is 0.57, which is better than predictions based on fixed specific thresholds with accuracy of 0.36.

Table 3. Confusion matrix for fire severity classification between FESM and predictions based on vegetation specific and fixed thresholds.

| Vegetation specific | | | | | Fixed | | | | |
|---|---|---|---|---|---|---|---|---|---|
| | Extreme | High | Moderate | Low | | Extreme | High | Moderate | Low |
| Extreme | 4345 | 2378 | 6 | 3 | Extreme | 1448 | 2822 | 2027 | 435 |
| High | 1490 | 6947 | 605 | 1 | High | 1430 | 3561 | 3358 | 694 |
| Moderate | 3 | 5702 | 9338 | 5 | Moderate | 998 | 4633 | 7084 | 2333 |
| Low | 0 | 172 | 7125 | 2372 | Low | 161 | 1722 | 5264 | 2522 |

To evaluate the model's performance in fire severity prediction, we apply the leave-one-year-out cross-validation method. We
validate the fire severity predictions against the observed burn severity derived from Landsat images and compare the
predictions based on the RF model with (and without) severity classification method. Figures 8 (a), (b) and (c) display the
scatterplots of fire severity prediction against fire severity observations based on RF model without severity classification,
with severity classification using the fixed threshold and using the vegetation-specific threshold, respectively. Arguably, the
predictions without severity classification show strong underestimation of high fire severity events and overestimation of low
burn severity events, with $R^2$ value of 0.62, RMSE and MAE are 0.14 and 0.11, respectively. The distributions of predictions
with severity classification using the fixed threshold do not agree well with observations, though showing higher $R^2$ (0.79),
lower RMSE and MAE values of 0.11 and 0.08, respectively. Predictions with severity classification using the vegetation-
specific threshold exhibit better fire severity prediction results for high-, moderate- and low-severity events with improved $R^2$,
RMSE and MAE, which are 0.89, 0.07 and 0.05, respectively.

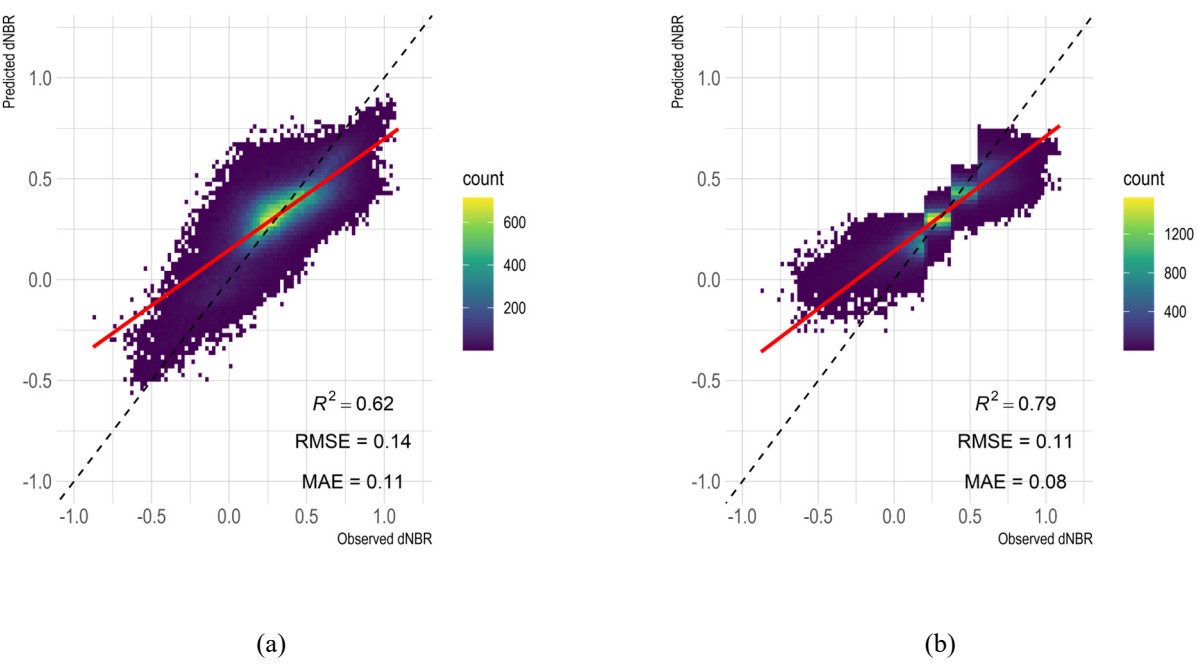

(a)                                                                                    (b)

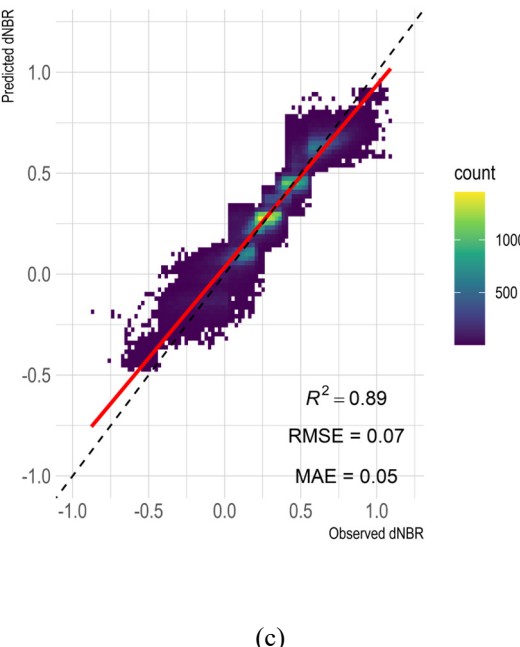

(c)

Figure 8. Scatterplots of fire severity prediction against observations based on XGBoost model (a) without severity classification; (b) with severity classification using the fixed threshold; and (c) with severity classification using the vegetation-specific threshold.


We also evaluate the model's ability of capturing the fire severity dynamics and magnitude in terms of mean fire severity for
the selected wildfires. Figure 9 (a) displays the dynamics of predicted fire severity based on RF model with and without severity
classification, while Figures 9 (b), (c) and (d) show the dynamics of associated performances of $R^2$, RMSE and MAE,
respectively. The predictions without severity classification are unable to capture the dynamics of mean fire severity, having
the lowest R2 and highest RMSE and MAE values. While the dynamics of the predicted fire severity with severity classification
has better correlation with the observed ones compared to those without severity classification, especially the results with
severity classification using the vegetation-specific threshold, which exhibit the best performance of predicting fire severity
magnitude with the largest $R^2$ and lowest RMSE and MAE values. These results indicate that severity classification is an
important process to improve the performance of fire severity prediction models.

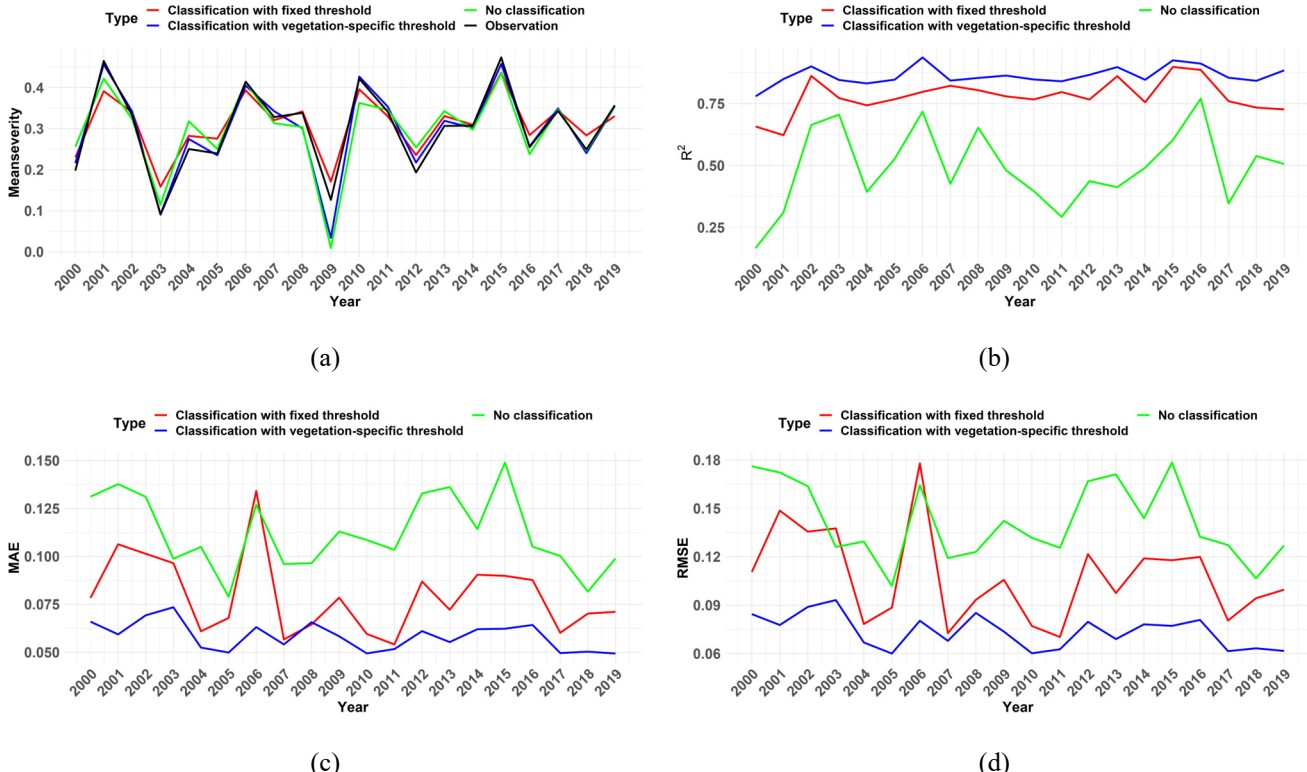

Figure 9. Time series of (a) mean fire severity, (b) $R^2$, (b) RMSE and (c) MAE from 2000 to 2019 based on XGBoost models without severity classification and with severity classification using the fixed and vegetation-specific threshold.


Figure 10 depicts a summary plot of estimated SHAP values coloured by the feature values, ranked from top to bottom by their
importance. It is shown that preNBR is the most important feature in the model, followed by FFDI. The KBDI is also crucial
in the model. The topographic factors are also contributing to the model. We can find that having a high preNBR is associated
with high and positive values on the model output, indicating the larger preNBR is the prerequisite of more severe wildfire.
Similar to the effect of preNBR on the model output, a high FFDI is always associated with high and positive SHAP values,
which means the more severe fire weather could lead to more destructive wildfires. Though some high KBDI is found to be
associated with negative SHAP values, the KBDI still shows strong positive effect on the model output, reflecting the fact that
the dry condition could favour the fire behaviour. Regarding the topography, the large slope and TPI tend to have positive
SHAP values, meaning the more severe fire tends to occur in steeper and higher position.

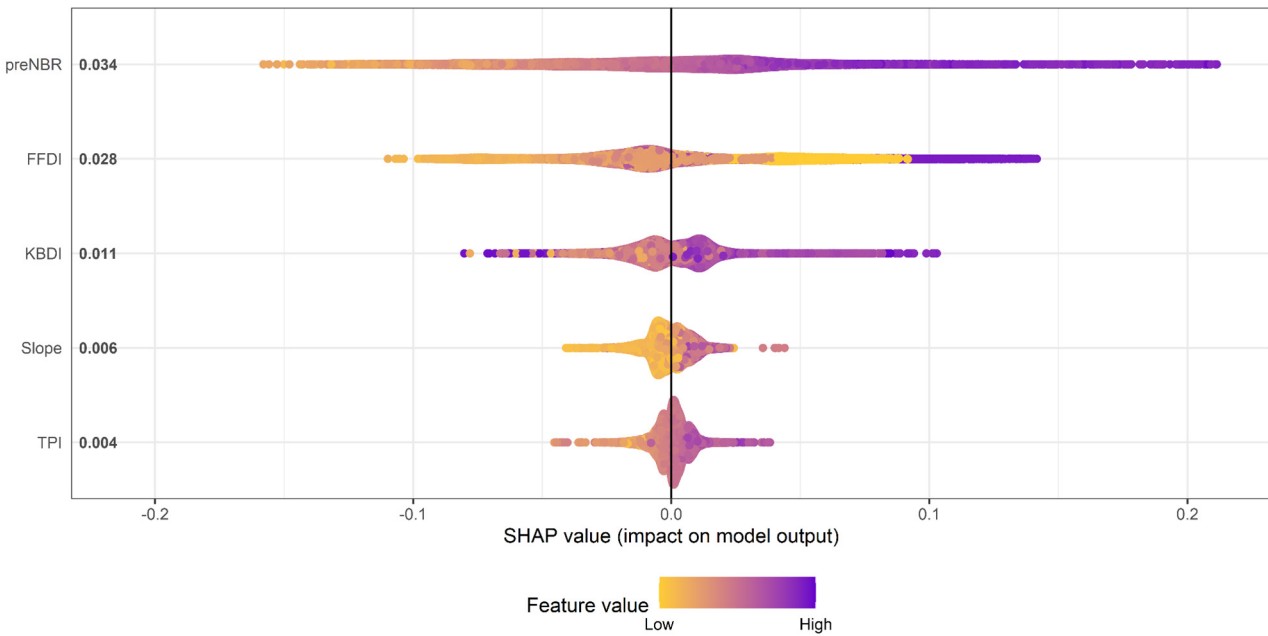

Figure 10. The SHAP values for variables predicting fire severity based on XGBoost model.


Fig. 11 displays the partial dependence plot (PDP) for each feature in the model. From Figure 11, it can be shown that the
preNBR has a strong positive association with the dNBR, implying that dNBR increases with the preNBR rapidly. The FFDI
shows a non-monotonic relationship with dNBR, with a decreasing trend observed when it is less than 30, a steady increasing
trend between 30 to 65 and significant increasing after it exceeds 65, suggesting that the fire weather dependence is more
complex. The weak correlation between KBDI and dNBR, within the range of KBDI lower than 400, indicates that KBDI has
nearly no influence when it is below 400. While the positive correlation between KBDI and dNBR, within the range of 400 to
600, suggest that the dry condition would intensify the fire severity. However, a declining trend of KBDI is found when it
exceeds 600, meaning the impact of KBDI on dNBR becomes weaker. Regarding the slope, a negative association with dNBR
is observed when it is below 3, while a positive relationship is found when it exceeds 3. The TPI shows an overall positive
association with dNBR. These findings demonstrate that fire severity tends to be higher on steeper slopes and in hilltops.

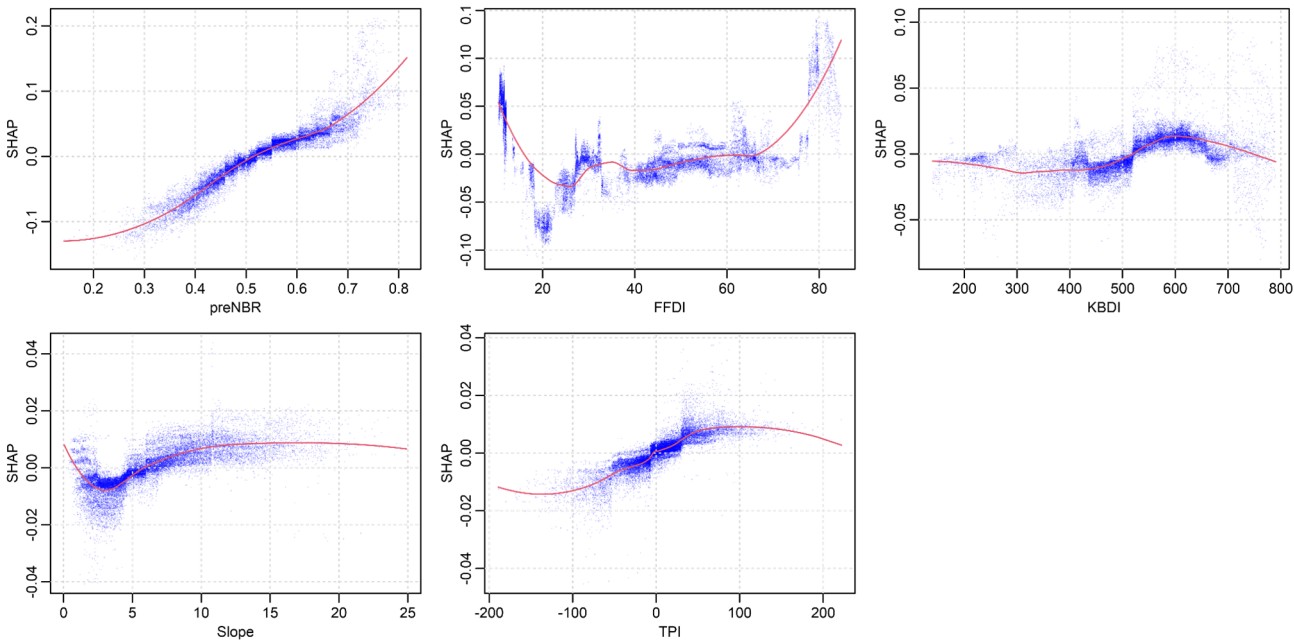

Figure 11. The variation of SHAP values as variables change.

**5 Discussion**

This study shows that the proposed predictive technique is capable of providing robust fire severity prediction information, which can be used for forecasting seasonal fire severity and, subsequently, impacts on biodiversity and ecosystems under future projected climate conditions.

We find that the RF is effective in classifying fire events into different levels of fire severity and XGBoost is a useful method to characterise the relationships between fire severity and explanatory variables (e.g., preNBR, FFDI, KBDI, slope and TPI). Fire severity is a complex function of explanatory variables gradients and these relationships may vary in different vegetation type and severity levels. The preNBR, an approximation of the pre-fire vegetation condition, plays an important role in classification and prediction, as the change in NBR pre- and post-fire, i.e. dNBR, will be dependent on both the condition of the vegetation before the fire and the degree of change to vegetation after the fire. The preNBR, indicating the pre-fire vegetation condition, might be related to the pre-fire drought. For example, drought reduces the water content of foliage (Choat et al. 2018), thus reducing preNBR, so the maximum absolute change in NBR (dNBR) possible might be smaller during a drought year than a non-drought year. The FFDI is found to be important in fire severity classification and prediction. The meteorological conditions are proven to be the most influential predictors in determining the magnitude of fire severity (Clarke et al., 2014; Bowman et al., 2021). The FFDI is the index of fire weather severity during the fire season thus is workable in determining the potential burn severity level. KBDI is another important variable in fire severity classification. It is known

that drought can create conditions that favour severe fires (Abram et al. 2021) and that the combined effects of fire and drought
can contribute to plant population declines (Gallagher et al. 2022; Nolan et al. 2021) and ecosystem transformation (Keith et
al. 2022). Severe drought conditions also directly contribute to forest flammability (Nolan et al. 2020). More importantly, the
frequency, intensity and duration of drought conditions are  projected to shift under future climates (Ukkola et al. 2020). These
changes in drought regimes will likely be associated with increases in the size, frequency and severity of fires (Abram et al.
2021). TPI and slope, as important topographic factors, also have considerable influence on low fire severity. For example,
Bradstock et al. (2010) found burn severity is lower in valleys, probably due to effects of wind protection and higher fuel
moisture in moderating fire behaviour. Barker et al. (2018) found that the probability of low severity increased with slope. In
this study, we find that fire severity tends to be higher on steeper slopes and higher position, this might be that steep slopes
can intensify fire behaviour by creating a chimney effect that draws in air and accelerates the fire (Andrews and Bradshaw,
2012; Jolly et al., 2015; Seginer and Brandl, 2007.). Besides, higher elevations generally have lower air pressure and reduced
humidity, which helps fire burn more intensely (Abatzoglou and Kolden, 2011; Holden et al., 2018). Additionally, vegetation
on steep slopes can be thicker and more continuous, providing more fuel for the fire (Collins et al., 2009; Pausas and Fernández-
Muñoz, 2012).
One limitation of this study is that it does not consider the vegetation vertical structure parameters in the fire severity model,
which have been shown to influence fire behavior. Agee (1996) showed that manipulating forest structure can help to reduce
the severity of fire events, e.g., by reducing the crown bulk density the high severity fire would be effectively limited. Fang et
al. (2015) evaluated the influences and relative importance of fire weather, topography, and vegetation structure on fire size
and fire severity, which showed fire weather was the dominant driving factor for fire size, while vegetation structure exerted
stronger influences on fire severity. The study by Fernández-Guisuraga et al. (2021) indicated that severe ecosystem damage
was mainly driven by vegetation structure rather than topography, for example high canopy density was the main driver of
high burn severity. Detailed and accurate vegetation structure data require extensive field inventory and thus are mostly
regionally restricted. With the development of Global Ecosystem Dynamics Investigation (GEDI) project, it is possible to
derive reliable forest vertical structure parameters from satellite with relatively high spatial resolution and global coverage
(Dubayah et al., 2020). An extension of this study should incorporate data from GEDI into the fire severity model, which
would represent an advancement in understanding and predicting the impact of wildfires. Besides, topographic data derived
from SRTM presents its limits, especially in vegetated areas and terrains with pronounced slopes or certain aspects
[Gorokhovich and Voustianiouk, 2006; Shortridge and Messina, 2011]. The advances in DEM technology, as evidenced by
the improvements in the SRTM data, such as SRTM-derived 1 Second -and 3 seconds- Digital Elevation Models Version 1.0
for Australia, and the introduction of global COPDEM30 and TanDEM-X data [Hawker et al., 2022], offer opportunities for
refining fire-topography relationship analyses and potentially providing more precise fire severity prediction results. The
introduction of vegetation specific thresholds is proven to be beneficial for fire severity classification. The range of dNBR
varies significantly with vegetation types, and thus applying a fixed threshold in dNBR would lead to a large amount of
misclassification in fire severity levels. This kind of misclassification error is mitigated by using vegetation specific thresholds
in dNBR. The vegetation type also plays an important role in the RF model. The relative influence of vegetation type is larger
than the topographic factors while the deviation of vegetation type is the largest in the meantime. The relative influence of
vegetation type and the deviation changes with the number of vegetation types and its fractions in the fire event. For example,
five vegetation types were affected in the 2002 wildfire, and the fractions of vegetation types are: dry sclerophyll forests
(shrubby subformation) (30%), grassy woodlands (31 %), wet sclerophyll forests (grassy subformation) (23%), dry sclerophyll
forests (Shrub/grass subformation) (14%) and grasslands (2%). While in the 2019 wildfire, seven vegetation types were
affected, dry sclerophyll forests (shrubby subformation) accounts for 92% of the burn area. The relative influence of vegetation
type in the 2002 wildfire is around 10% while only 5% in the 2019 wildfire. This could also explain why no significant
differences are found between fire severity maps using vegetation specific thresholds and fixed thresholds in the 2019 wildfire.
Since more than 90% of the burn area in the 2019 wildfire is covered by dry sclerophyll forests (shrubby subformation) and
the fixed thresholds are adopted from the thresholds of dry sclerophyll forests (shrubby subformation), the fire severity
classification for 2019 wildfire is almost equal to the fire severity classification for dry sclerophyll forests (shrubby
subformation).
This study develops a predictive technique which is capable of providing robust fire severity classification and prediction
information for historical events, which also has the potential to forecast the seasonal fire severity. The input variables to the
model could be obtained from other forecast models: fire weather related variables can be extracted from the Weather Research
and Forecasting (WRF) model. The NBR images are derived from the Landsat 5,7 and 8 in this study, while it is also applicable
to other image sources based on the reflectance information form NIR and SWIR, such as the new launched Landsat 9 and
Sentinel-2 (Mallinis et al., 2018; Howe et al. 2022). Owing to the seasonality characteristic of preNBR, we can infer the
preNBR in the fire season based on the historical preNBR time series derived from the image sources.The vegetation type and
topographic factors are static variables, while the variables for calculating FFDI and KBDI, e.g., wind speed, relative humidity,
precipitation, air temperature, are available from WRF outputs. Quick assessment of fire severity for wildfires are accessible
based on the proposed predictive technique, once the burn area are derived from the burn area prediction models (Alkhatib,
2014: Castelli et al., 2015) or monitoring products (e.g., MODIS Burned Area Product, MCD64A1).
With the rapid development of new technologies such as LiDAR and Unmanned Aerial Vehicle (UAV), integration of data
from these platforms can represent a promising avenue to enhance our understanding and management of wildfires. LiDAR
technology, with its capability to produce high-resolution vegetation structural and topography information could facilitate the
accurate modelling of fire severity (Hudak et al., 2012; Hébert et al., 2017). On the other hand, the agility and precision of
UAVs in data collection enable real-time monitoring of fire spreading, which significantly enhances our ability to map burn
areas in real-time (Véga et al., 2018; Zheng et al., 2019).

## 6 Conclusions

This study introduces the vegetation specific thresholds in fire severity classification for wildfires over NSW, Australia. We use the pre-fire season drought conditions, topography, and the fire season meteorological conditions as input to build the predictive model and the performances are validated by EXtreme Gradient Boosting (XGBoost) to predict the fire severity, proxied by dNBR.

Using the vegetation specific thresholds we could improve the classification accuracy in fire severity levels. Specifically, compared with the fire severity classifications from FESM over NSW, we found fire severity classification results using vegetation specific thresholds show good agreement to those from FESM, with accuracy of 0.64 and 0.76 in extreme and high severity classification. Using a leave-one-out cross-validation, the severity classification results showed an improved classification accuracy of 0.75 based on the proposed vegetation specific thresholds, compared to those based on fixed thresholds (0.69). The predictive performance of XGBoost model is improved as well based on the classification results, with determination coefficient ($R^2$), mean absolute error (MPE) and root mean square error (RMSE) values of 0.89, 0.05, and 0.07, respectively. We show that the preNBR is the most important variable in fire severity classification and prediction, followed by FFDI and KBDI. The PDP of FFDI and KBDI indicate that the likelihood of high severity increases when weather and drought conditions become more severe. From the responses of dNBR to topographic factors, the probability of high severity increases with slope and elevation. The role of vegetation type in fire severity prediction becomes more important for large fires where more diverse vegetation is affected.

The results demonstrate that the prediction technique performs well predicting fire severity of historic fires (2000-2019) in the Australian state of NSW, while it also shows the potential to be applicable for seasonal fire severity forecasts, owing to the availability of the predictor variables in seasonal forecasting outputs. With the expected increase in wind speed, temperature and drought conditions exhibited in future climate projections, this prediction technique can also be used to evaluate the variation of fire severity under climate change. Future challenges of this study include incorporating different variables, such as refined topography as well as weather and vegetation structure, from various data source to improve the accuracy of fire severity prediction and scaling up the application of the developed model globally. In addition, the sensitivity analysis of the selected time window to define the fire event and obtain the associated weather conditions is promoted to improve our understanding of the relationship between weather conditions and fire occurrences. By adjusting the time window and possibly integrating more precise burn date data, we can work towards a more accurate and physically meaningful analysis of fire events and their contributing factors.

**Author contributions:** Kang He: Data curation, Visualization, Writing-Original draft preparation. Xinyi Shen: Supervision, Methodology, Writing- Reviewing and Editing. Emmanouil N. Anagnostou: Supervision, Methodology, Writing-Reviewing and Editing Cory Merow: Methodology, Writing-Reviewing and Editing. Efthymios Nikolopoulos: Data curation, Writing-

Reviewing and Editing. Rachael Gallagher: Data curation, Writing-Reviewing and Editing Feifei Yang: Methodology,
Writing- Reviewing and Editing.

**Competing interests:** The contact author has declared that none of the authors has any competing interests.

**Acknowledgements:** This research was supported by National Science Foundation HDR award entitled "Collaborative
Research: Near term forecast of Global Plant Distribution Community Structure, and Ecosystem Function". Kang He received
the support of China Scholarship Council for four years' Ph.D. study in University of Connecticut (under grant agreement no.
494 201906320068).

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
