# Peer review of "Improving fire severity prediction in south-eastern Australia using"

_Natural Hazards and Earth System Sciences, 2023_

## Author Comment (AC1)

Dear authors, your work focused on the possibility of improving fire severity prediction through specific vegetation information and indexes in a wildfire-affected area in south-eastern Australia. The work is generally well written and I found it interesting.

**Respond**: We appreciate the reviewer's constructive comments on the manuscript to further improve the quality and the contribution of our work. Below are the authors' responses on all of the reviewer's questions and suggestions. The reviewer's comments are marked as **red**, while our responses are marked as **blue**.

In any case, different issues need to be considered in your revision:

- I have a first comment about the main focus on fire severity that characterizes your research: fire behavior (that is also described by fire severity), also depends on several other factors that jointly influence it over time. In particular, even if a brief discussion about it is presented in lines 355-363, I suggest better clarifying this issue, especially explaining the relevance of considering all these factors together in fire behavior analysis. For instance, no reference to the importance of the vertical structure of forested areas (DBH, Canopy Cover, CBH, CBD) in this kind of analysis is proposed in the manuscript. Please improve the respective section of the paper by looking at these suggestions.

Respond: Thanks for the suggestions. We realize that vegetation structure can play an important role in fire behavior and it is a limitation of this study that did not include vegetation structure in the fire severity model. We consider this a future development based on some recent satellite data on vegetation height, which can extend the application of this model. We have added discussion in the revised paper regarding this point.

From line 416 to 427 in the revised manuscript:

"One limitation of this study is that it does not consider the vegetation vertical structure parameters in the fire severity model, which have been shown to influence fire behavior. Agee (1996) showed that manipulating forest structure can help to reduce the severity of fire events, e.g., by reducing the crown bulk density the high severity fire would be effectively limited. Fang et al. (2015) evaluated the influences and relative importance of fire weather, topography, and vegetation structure on fire size and fire severity, which showed fire weather was the dominant driving factor for fire size, while vegetation structure exerted stronger influences on fire severity. The study by Fernández-Guisuraga et al. (2021) indicated that severe ecosystem damage was mainly driven by vegetation structure rather than topography, for example high canopy density was the main driver of high burn severity. Detailed and accurate vegetation structure data require extensive field inventory and thus are mostly regionally restricted. With the development of Global Ecosystem Dynamics Investigation (GEDI) project, it is possible to derive reliable forest vertical structure parameters from satellite with relatively high spatial resolution and global coverage (Dubayah et al., 2020). An extension of this study should incorporate

data from GEDI into the fire severity model, which would represent an advancement in understanding and predicting the impact of wildfires."

Agee, James K. (1996). "The influence of forest structure on fire behavior." In Proceedings of the 17th annual forest vegetation management conference, pp. 52-68.

Fang, L., Yang, J., Zu, J., Li, G. and Zhang, J., (2015). Quantifying influences and relative importance of fire weather, topography, and vegetation on fire size and fire severity in a Chinese boreal forest landscape. Forest Ecology and Management, 356, pp.2-12.

Fernández-Guisuraga, J.M., Suárez-Seoane, S., García-Llamas, P. and Calvo, L., (2021). Vegetation structure parameters determine high burn severity likelihood in different ecosystem types: A case study in a burned Mediterranean landscape. Journal of environmental management, 288, p.112462.

Dubayah, R., Blair, J.B., Goetz, S., Fatoyinbo, L., Hansen, M., Healey, S., Hofton, M., Hurtt, G., Kellner, J., Luthcke, S. and Armston, J., 2020. The Global Ecosystem Dynamics Investigation: High-resolution laser ranging of the Earth's forests and topography. Science of remote sensing, 1, p.100002.

- Lines 114: here you mention the use of Sentinel 2 together with Landsat 8 data in obtaining pre-NBR. Why did you use both and how you considered the different resolutions of band products in your analysis is not clear or evidenced. Please clarify it by adding an explanation in the methodology section, specifying what satellite data you considered, when, and why also considering the post-processing procedure followed in L8 /S2 data-elaboration. In this regard, you should also improve the Discussion by focusing on other research based on satellite data processing and use in fire-behavior analysis.

Respond: To pre-process NBR data, we apply a cloud- and snow-masking algorithm to remove any snow, clouds, and their shadows from all Landsat imagery. Therefore, there will be many blank pixels with NaN value within the fire boundary. To fill the gaps, we adopt the pixel value from the Sentinel-2 image available in the same period. We have added the steps on how to obtain the dNBR image.

From line 112 to line 120 in the revised manuscript:

"The calculation of a dNBR-image is described as follows: (1) determine an individual fire from NPWS Fire History; (2) collect the most recent Landsat images based on the tags demarcating the start and end times of each individual fire; (3) apply a cloud- and snow-masking algorithm to remove snow, clouds, and their shadows from all imagery based on each sensor's pixel quality assessment band; (4) use the auxiliary satellite images (e.g., Sentinel-2) to fill the blank pixels in the cloud-free images from step (3) to obtain the pre and post NBR composites; (5) subtract pre- and post-NBR images to create a dNBR composite with the smallest possible cloud and shadow extent. The

dNBR typically ranges from -2 to +2, with high positive values indicating severe burn damage where the vegetation has been completely consumed. Values around zero suggest either unburned areas or areas where the fire had a very low impact. Negative values can indicate an increase in vegetation, which might be due to vegetation recovery over time or errors in the analysis."

- Line 168: why did you choose to consider *20* subsets of fire samples? Please justify this choice.

Respond: The reason we have 20 subsets of fire samples is that we derived the dNBR and the associated variables from the largest wildfire of each year from 2000 to 2019. In this way, we keep the balance between the sample size and the sample representative in the model.

From line 195 to line 197 in the revised manuscript:

"The fire samples from 2000 to 2019 are firstly divided into 20 subsets depending on the year the fire occurred, and this holdout method is repeated 20 times. Each subset represents the samples from the wildfire with the largest burn area in the corresponding year."

- Lines 41-54 should be moved to Discussion, where a comparison between your work and other research is needed looking at your paper outline and workflow.

Respond: Thanks for this suggestion. After discussing with the coauthors, we think we are doing the literature review in this paragraph. So we will keep these sentences in the introduction section.

- Please improve the final part of the Discussion citing the possibility to use also different data and tools (such as LiDAR or UAV-based multi-spectral data) in forest fire behavior analysis.

Respond: We have added a paragraph emphasizing the application of LiDAR and UAV in forest fire management in the revised paper.

From line 460 to line 465 in the revised manuscript:

"With the rapid development of new technologies such as LiDAR and Unmanned Aerial Vehicle (UAV), integration of data from these platforms can represent a promising avenue to enhance our understanding and management of wildfires. LiDAR technology, with its capability to produce high-resolution vegetation structural and topography information could facilitate the accurate modelling of fire severity (Hudak

et al., 2012; Hébert et al., 2017). On the other hand, the agility and precision of UAVs in data collection enable real-time monitoring of fire spreading, which significantly enhances our ability to map burn areas in real-time (Véga et al., 2018; Zheng et al., 2019). "

Hudak, A. T., Strand, E. K., Vierling, L. A., Byrne, J. C., Eitel, J. U., & Martinuzzi, S. 2012. Quantifying aboveground forest carbon pools and fluxes from repeat LiDAR surveys. Remote Sensing of Environment, 123, 25-40.

Hébert, F., & Mallet, C. 2017. Forest fire severity assessment using LiDAR in a Mediterranean environment. Remote Sensing, 9(9), 908.

Véga, C., Martín, M. P., López, F. J., García, A. M., & Pérez, J. A. (2018). Fire spread and vegetation monitoring by using a UAV system. Drones, 2(4), 31.

Zheng, D., Jiang, Y., & Cheng, T. (2019). UAV-based remote sensing technology in the rapid monitoring of forest fires. International Journal of Remote Sensing, 40(11), 4257-4275.

- There are no clear pieces of evidence about future challenges starting from your research. Please enrich the Conclusion in this regard.

Respond:  We have added sentences addressing the future challenges of the study.

From line 487 to line 489 in the revised manuscript:

"Future challenges of this study include incorporating different variables, such as refined topography as well as weather and vegetation structure, from various data source to improve the accuracy of fire severity prediction, and scaling up the application of the developed model globally."

Other minor comments are reported below:

- line 17: what did you mean by "fire weather"? please clarify

Respond: The fire weather means the weather condition during the fire season, like wind speed, air temperature, humidity. We have clarified it in the revised paper.

In line 17 in the revised manuscript:

"which is further used to predict fire severity using antecedent drought conditions, fire weather (i.e., wind speed, air temperature and atmospheric humidity), and topography of the fire season (November to March)."

- line 17: "topography *during* the fire season". Specify the duration of the fire season and add a reference (what months were considered as fire season?)

Respond: Fire season in Australia refers to the period of the year when wildfires, also known as bushfires in Australia. The fire season in the southern parts of the country, including regions such as New South Wales, Victoria, South Australia, and Tasmania, generally peaks during the warmer months, from late spring through to early autumn (approximately November to March). This is when the vegetation has dried out, and hot, dry, and often windy conditions prevail, making it easier for fires to start and spread rapidly.

From line 59 to line 62 in the revised manuscript:

"One such region is the southeast coast of Australia which is subject to annual fire seasons (from November to March, Collins et al., 2022) vary in extent and severity and has a high richness of endemic plant species adapted to particular fire regimes (Gallagher et al., 2021)."

Collins, L., Clarke, H., Clarke, M.F., McColl Gausden, S.C., Nolan, R.H., Penman, T. and Bradstock, R., 2022. Warmer and drier conditions have increased the potential for large and severe fire seasons across south-eastern Australia. Global Ecology and Biogeography, 31(10), pp.1933-1948.

Gallagher, R. V., Allen, S., Mackenzie, B. D., Yates, C. J., Gosper, C. R., Keith, D. A., ... & Auld, T. D. (2021). High fire frequency and the impact of the 2019–2020 megafires on Australian plant diversity. Diversity and Distributions, 27(7), 1166-1179.

- line 22: "forecasting /forecast" repetition. Please change one term

Respond: We use forecast throughout the paper.

- line 40: add a reference

Respond: A reference has been added for dNBR

Keeley, J.E., 2009. Fire intensity, fire severity and burn severity: a brief review and suggested usage. International journal of wildland fire, 18(1), pp.116-126.

- Figure 1: increase the size of the legend. Is also not clear if colors are only related to the years or also depends on fire extension (since polygons in the figure are different colored but have also different size). Please specify

Respond: We have redesigned the figure to make it clearer.

173

[Figure]

Figure 1. Locations of study wildfires over New South Wales (NSW), Australia. The
burn area is from NSW National Parks and Wildlife Service (NPWS) Fire History –
Wildfire and Prescribed Burns dataset.

- line 95, eq.1: add a reference about dNBR equation

Respond: A reference has been added.

Keeley, J.E., 2009. Fire intensity, fire severity and burn severity: a brief review and
suggested usage. International journal of wildland fire, 18(1), pp.116-126.

- line 119: is there a repetition of "DEM"? Please clarify since is not clear

Respond: We apologize for the mistake, we have removed the repetition of DEM.

- line 124: "wildfire environment": what did you mean with "environment"? Please
  clarify and rephrase the sentence

Respond: We apologize for the confusion, we have rewritten this sentence.

"In addition to fuels and terrain, weather is another important factor in wildfires."

- lines 206-213 and line 221: change "figure 2" with "figure 3"

Respond: We have revised it accordingly.

- Figure 3: increase the size of legends

Respond: We have revised it accordingly.

[Figure]

191

192    • line 223: add space "were_collected"

193    Respond: We have revised it accordingly.

194    • line 231: "Note that" seems quite colloquial, why not change it with something like "is
195    important to consider that" or similar?

196    Respond: Thanks for the suggestion. We have changed this sentence to

197    "It is important to be aware that the classification step is merely used to improve the
198    consecutive regression accuracy, rather than the final severity categorization result"

199    • lines 227-231: is not clear how the different percentages were adopted

200    Respond: We have clarified this in the method section.

201    From line 162 to line 167 in the revised manuscript:

202    "The dNBR of all burnt pixels for each vegetation type are collected and a set of dNBR
203    values at the quantiles varying from 5% to 35% representing the threshold for low
204    severity classification, quantiles varying from 35% to 65% representing the threshold
205    for moderate severity classification, and quantiles varying from 65% to 95%
206    representing the threshold for high severity classification. For example, a classified burn
207    severity sample can be obtained using the thresholds for high, moderate, and low severity
208    at 85% quantile, 55% quantile and 25% quantile, respectively."

209    • Figure 4: legends and descriptions are too small

210    Respond: We have increased the size accordingly.

[Figure]

211

- Figure 5: as Figure 4

  Respond: We have revised it accordingly.

- Figure 6: remove the term "The" in the caption

  Respond: We have revised it accordingly.

- Figure 9: legends and items are too small

  Respond: We have increased the size accordingly.

[Figure]

218

- lines 338-339: repetition of "method", please rephrase

  Respond: We have removed the repetition word.

- line 366: "mis-classification" or "misclassification"?

222    Respond: It should be "misclassification"

223   ● line 370: add space: "the_2002"

224    Respond: We have revised it accordingly.

225

226 Good work and best regards

227

---

## Author Comment (AC2)

This paper proposes a novel approach for fire severity, with a focus on the escalating wildfire activity in southern Australia. By introducing a vegetation-type specific fire severity classification method applied to satellite imagery, the paper lays the groundwork for more accurate prediction and assessment of wildfire impacts on ecosystems. The paper is well written and organized, but there are few items that could be addressed to strengthen the importance of the work.

**Respond**: We appreciate the reviewer's constructive comments on the manuscript to further improve the quality and the contribution of our work. Below are the authors' responses on all of the reviewer's questions and suggestions. The reviewer's comments are marked as **red**, while our responses are marked as **blue**.

Introduction

The authors state that no classification scheme for southern Australia exists, however literature showed works towards this, see for example (Collins et al., 2018; Dixon et al., 2022; Gale et al., 2023; Gibson et al., 2020). There are also accessible datasets on fire severity available from other sources, for the country, https://datasets.seed.nsw.gov.au/dataset/fire-extent-and-severity-mapping-fesm

**Respond**: We are sorry didn't state this sentence clearly. While most fire severity classifications are based on the field assessed index, like Composite Burn Index (CBI), and interpretation from aerial photographs, which are always labor intensive and time consuming, especially for large regions. And those prediction models rely on establishing the relationships between satellite-derived index (dNBR) and CBI or appearances from aerial photographs.

Our study tried to propose a more straight dNBR-based fire severity classification scheme based on the statistical analysis of dNBR for historical wildfire events, without relying on the CBI or aerial photographs.

From line 63 to line 72 in the revised manuscript:

"The most prevailing fire severity classification scheme mainly rely on the in-situ measurements of Composite Burn Index (CBI, Key and Benson, 2006; Lutes et al., 2006) and aerial photographs identification (Collins et al., 2018; Dixon et al., 2022) which are available for certain regions and for limited vegetation types under certain climate (Eidenshink et al., 2007; Keeley et al., 2009; Tran et al., 2018). However, obtaining CBI and interpreting aerial photographs are always labor-intensive and time-consuming, especially over large areas, while inferring fire severity levels directly from satellite-derived dNBR is more efficient for large-scale applications, yet no dNBR-based fire severity classification scheme has been proposed for regions such as the southeast coast of Australia, which is subject to annual wildfire seasons and varies greatly in vegetation types with high richness of endemic plant species adapted to particular fire regimes (Gallagher et al., 2021)"

References:

Key, C.H. and Benson, N.C., 2006. Landscape assessment (LA). FIREMON: Fire effects monitoring and inventory system, 164, pp.LA-1.

Lutes, D.C., Keane, R.E., Caratti, J.F., Key, C.H., Benson, N.C., Sutherland, S. and Gangi, L.J., 2006. FIREMON: Fire effects monitoring and inventory system. Gen. Tech. Rep. RMRS-GTR-164. Fort Collins, CO: US Department of Agriculture, Forest Service, Rocky Mountain Research Station. 1 CD., 164.

Collins, L., Griffioen, P., Newell, G., Mellor, A., 2018. The utility of Random Forests for wildfire severity mapping. Remote Sensing of Environment 216, 374–384. https://doi.org/10.1016/j.rse.2018.07.005

Dixon, D.J., Callow, J.N., Duncan, J.M.A., Setterfield, S.A., Pauli, N., 2022. Regional-scale fire severity mapping of Eucalyptus forests with the Landsat archive. Remote Sensing of Environment 270, 112863. https://doi.org/10.1016/j.rse.2021.112863

Eidenshink, J., Schwind, B., Brewer, K., Zhu, Z.L., Quayle, B. and Howard, S., 2007. A project for monitoring trends in burn severity. Fire ecology, 3(1), pp.3-21.

Keeley, J. E. (2009). Fire intensity, fire severity and burn severity: a brief review and suggested usage. International journal of wildland fire, 18(1), 116-126.

Tran, B.N., Tanase, M.A., Bennett, L.T. and Aponte, C., 2018. Evaluation of spectral indices for assessing fire severity in Australian temperate forests. Remote sensing, 10(11), p.1680.

Fire severity:

As the technique for dNBR relies on NIR and SWIR, would it be possible to apply the proposed methods to other imagery sources, such as Sentinel or the new Landsat missions? If applicable, it would be beneficial to highlight this point as well for researcher wanting to apply the proposed approach.

**Respond:** Yes, this technique is applicable to other imagery source, with the correct band settings for NIR and SWIR.

From line 105 to line 108 in the revised manuscript,

"NBR can be computed by the Thematic Mapper (TM) and Enhanced Thematic Mapper Plus (ETM+) sensors on using Band 7 as the short-wave infrared (SWIR) and Band 4 for Landsat 4-7 and Band 5 for Landsat 8 as the near infrared (NIR) reflectance, respectively. While in Sentinel-2, SWIR and NIR are represented by Band 8 and Band 12, respectively."

And from line 451 to line 453 in the revised manuscript:

"The NBR images are derived from the Landsat 5,7 and 8 in this study, while it is also applicable
to other image sources based on the reflectance information form NIR and SWIR, such as the
new launched Landsat 9 and Sentinel-2 (Mallinis et al., 2018; Howe et al. 2022)."

References:

Mallinis, G., Mitsopoulos, I. and Chrysafi, I. Evaluating and comparing Sentinel 2A and
Landsat-8 Operational Land Imager (OLI) spectral indices for estimating fire severity in a
Mediterranean pine ecosystem of Greece. GIsci Remote Sens, 55(1), 1-18,
https://doi.org/10.1080/15481603.2017.1354803, 2018.

Howe, A.A., Parks, S.A., Harvey, B.J., Saberi, S.J., Lutz, J.A. and Yocom, L.L. Comparing
Sentinel-2 and Landsat 8 for burn severity mapping in Western North America. Remote Sensing,
14(20), 5249, https://doi.org/10.3390/rs14205249, 2022.

Topography:

The authors consider the SRTM as main DEM source, and in the discussion, they highlight how
topography appears as an important variable in their model. SRTM however presents limits,
especially in areas covered by vegetation, and in general, its error values have strong correlation
with terrain slope and certain aspect values (See e.g. (Gorokhovich and Voustianiouk, 2006;
Shortridge and Messina, 2011).

For Australia specifically, there is the availability of an upgraded SRTM [SRTM-derived 1
Second -and 3 seconds- Digital Elevation Models Version 1.0, which are an improved DEM
compared to the original SRTM.  Literature also highlighted that COPDEM30, and the
underlying TanDEM-X data, as the most recent and accurate global DEM, and (Hawker et al.,
2022) provided a further cleaned version of such a DEM without buildings and Vegetation. Did
the authors consider using this upgraded terrain information for the model?

**Respond:** Thank you for bringing to attention the limitations of SRTM data, especially in
vegetated areas and terrains with pronounced slopes or certain aspects. The points raised about the
correlation of SRTM error values with terrain characteristics, and the availability of improved
DEM sources such as the upgraded SRTM for Australia and the COPDEM30, are indeed very
pertinent.

We compared the original SRTM used in this study with the upgraded SRTM [SRTM-derived 1
Second Digital Elevation Models Version 1.0] for Australia, over the burn area from 2000 to 2019.
The results, as Figure 1 (a) shown in the response letter, indicate that the original SRTM and the
upgraded SRTM present similar spatial patterns in terms of the elevation over the burn area. We
also calculated the relative differences between the elevation from original SRTM and the
upgraded SRTM to the elevation from the upgraded SRTM, e.g. relative differences =
100*(original SRTM - upgraded SRTM)/ upgraded SRTM and present the result as Figure 1 (b)
in the response letter. We find that most of the difference range from -10 % to 10 %, which is not
the markable difference.

While this study mainly focuses on proposing a vegetation specific classification method to
improve the performance of fire severity prediction model, we acknowledge the potential benefits
of incorporating more refined elevation data to enhance the accuracy of our model, yet did not
utilize the upgraded SRTM or the cleaned version of COPDEM30 in our present analysis.
However, the prospect of applying these more accurate DEM sources is an exciting direction for
our future research endeavors.

[Figure]

(a)                                                                                      (b)

Figure 1. (a) Spatial patterns of elevation from original SRTM and the SRTM-derived 1 Second
Digital Elevation Models Version 1.0 and (b) the distribution of relative difference between
DEM from original SRTM and the SRTM-derived 1 Second Digital Elevation Models Version
1.0, over burn area from 2000 to 2019 in NSW;

From line 428 to line 431 in the revised manuscript:

"The advances in DEM technology, as evidenced by the improvements in the SRTM data, such as
SRTM-derived 1 Second -and 3 seconds- Digital Elevation Models Version 1.0 for Australia, and
the introduction of global COPDEM30 and TanDEM-X data [Hawker et al., 2022], offer
opportunities for refining fire-topography relationship analyses and potentially providing more
precise fire severity prediction results."

Weather:

How was the '1 day window' decided to get the weather event? Is there a physical meaning linked to this choice or was it operationally decided? I am not sure if it is possible, but have the authors investigated the sensitivity of the results to this window? Literature reported a known potential limitation of the fire history database as the fact that the date of the fire attribute does not always represent the exact burn date (Dixon et al., 2022). Dixon for example proposed a semi-automatic MODIS date-adjustment method to obtain the start and end fire dates: have the authors considered something similar?

**Respond:** In this study, the daily FFDI value for the day prior to the start of the wildfires is used as the input variable in the model. We use daily FFDI because FFDI is typically calculated on a daily basis, indicated by Australian Bureau of Meteorology (BoM, http://www.bom.gov.au/climate/maps/averages/ffdi/). This daily calculation allows for the assessment of fire danger to reflect current weather conditions, including temperature, humidity, wind speed, and recent rainfall, which are critical for determining the day-to-day fire risk.

We use the daily FFDI for the day prior to the start of the wildfires because we found that extreme values of the FFDI appeared at times close to the start of the wildfires, as presented by Figure 22, Figure 26, Figure 30, Figure 34, Figure 43 in Dowdy et al. (2009). The physical rationale behind this choice is rooted in the understanding that weather conditions can change rapidly and have immediate effects on fire behavior. Using the most potential extreme FFDI, indicating the extreme weather conditions, in the period leading up to a wildfire could address the impact of weather on wildfire risk.

From line 154 to line 158 in the reviser manuscript,

"The daily FFDI and KBDI values for the day prior to the start of the wildfires are used as the predictors in predicting burn severity, owing to the strong correlation in time between extreme values of the FFDI and the start of the wildfires [Dowdy et al., 2009]Using the most potential extreme FFDI, indicating the extreme weather conditions, in the period leading up to a wildfire could address the impact of weather on wildfire risk."

References:

Dowdy, A.J., Mills, G.A., Finkele, K. and De Groot, W., 2009. Australian fire weather as represented by the McArthur forest fire danger index and the Canadian forest fire weather index (p. 91). Melbourne: Centre for Australian Weather and Climate Research.

Regarding the sensitivity of the results to the selected time window, we have not yet conducted an extensive sensitivity analysis. Future research could explore varying the window of observation to assess the impact on model results and address the issue raised by Dixon et al. (2022). The burn area and the associated burn date data are from NPWS Fire History - Wildfires and Prescribed Burns Dataset (https://datasets.seed.nsw.gov.au/dataset/fire-history-wildfires-and-prescribed-burns-1e8b6), which we think has good data quality preserved by NSW Department of Climate Change, Energy, the Environment and Water.

From line 492 to 494 in the revised manuscript:

"In addition, the sensitivity analysis of the selected time window to define the fire event and obtain the associated weather conditions is promoted to improve our understanding of the relationship between weather conditions and fire occurrences. By adjusting the time window and possibly integrating more precise burn date data, we can work towards a more accurate and physically meaningful analysis of fire events and their contributing factors."

Fire severity classes:

As it is my understanding, the severity is based on the dNBR which ranges from -n to +n. Is there a meaningful range of this value representing the severity? (I assume the higher in the positive, the higher the expected impact of the fire -if this is the case, please can you clarify it for the readers not too familiar with the approach? When selecting the quantiles, does the author use the full range of dNBR or focus on a selected part of the distribution (would that matter, if that's the case?).

**Respond:** The differenced Normalized Burn Ratio (dNBR) is a metric used to quantify burn severity by analyzing the difference in the spectral signature of an area before and after a fire event. The dNBR is calculated by subtracting the post-fire NBR from the pre-fire NBR, resulting in values that theoretically range from -2 to +2. The scale of dNBR values indeed reflects the severity of a fire with high positive values indicate severe burn damage where the vegetation has been completely consumed. Values around zero suggest either unburned areas or areas where the fire had a very low impact. Negative values can indicate an increase in vegetation, which might be due to vegetation recovery over time or errors in the analysis.

From line 117 to line 120 in the revised manuscript:

"The dNBR typically ranges from -2 to +2, with high positive values indicate severe burn damage where the vegetation has been completely consumed. Values around zero suggest either unburned areas or areas where the fire had a very low impact. Negative values can indicate an increase in vegetation, which might be due to vegetation recovery over time or errors in the analysis."

In selecting the quantiles for analysis, the full range of dNBR values is generally considered to capture the complete spectrum of burn severity, the results will provide a comprehensive overview of all fire severities. In the context of our study, we have utilized the full range of dNBR values to ensure a broad assessment of fire severity across the landscape. This inclusive approach allows us to capture all degrees of burn severity, from low to extreme, offering a complete view of the fire's impact.

I find it a bit confusing that the methods describe a threshold selection, but the whole approach is clarified better in the discussion of the results at chapter 4.2. Would it be possible to restructure a bit this chapter in the method, to clarify how the selection is done?

**Respond:** Thanks for the suggestion. We have rewritten the method section to better clarify how to use the quantile based threshold in burn severity classification.

From line 161 o line 165 in the revised manuscript,

"The dNBR of all burnt pixels for each vegetation type are collected and a set of dNBR values at the quantiles varying from 5% to 35% representing the threshold for low severity classification, quantiles varying from 35% to 65% representing the threshold for moderate severity classification, and quantiles varying from 65% to 95% representing the threshold for high severity classification. For example, a classified burn severity sample can be obtained using the thresholds for high, moderate and low severity at 85% quantile, 55% quantile and 25% quantile, respectively."

Maybe this comes from my misinterpretation of the result chapter, but my understanding is that the ground truth for the severity is the 'observed severity' from Landsat for some specific fires (Figure 7). If this is the case, and the severity level is defined by a 'moving' threshold which in turn is defined by the best model in the training phase, how do you objectively define if the severity is 'under' or 'over' estimated as compared to the reality of the events? The observed severity is defined using a threshold derived from a 'training' of the model.

Would it be possible to compare your severity to some data independent from the threshold choice? I see for example for Australia some other datasets are available, such as https://data.gov.au/dataset/ds-nsw-c28a6aa8-a7ce-4181-8ed1-fd221dfcefc8/details?q=

**Respond:** Thanks for the suggestion. In the revised manuscript, we have used the fire severity classification maps from the Fire Extent and Severity Mapping (FESM) preserved by NSW Department of Climate Change, Energy, the Environment and Water as the independent source to validate the burn severity prediction maps from the model in this study.

From line 318 to line 339 in the revised manuscript:

"Figure 7 displays the fire severity maps for the 2016, 2017, 2018 and 2019 wildfires in NSW from FESM, along with predictions based on vegetation specific and fixed thresholds. For the wildfire in 2016, predictions based on vegetation specific thresholds show similar spatial patterns of fire severity to those from FESM, while predictions based on fixed thresholds significantly underestimate the fire severity in the high and extreme fire severity areas of the FSEM. Similarly for the wildfire in 2018, predictions based on fixed thresholds significantly underestimate high and extreme severity compared to the FESM map, while predictions based on vegetation specific thresholds slightly underestimate extreme severity. For the wildfire in 2017, both the FESM and predictions display similar spatial distributions of fire severity level with predictions based on fixed thresholds presents more low severity compared to FESM map. For the wildfire in 2019, however, predictions based on fixed thresholds tend to overestimate the fire severity as extreme in regions found to be high severity in FESM map, while predictions based on vegetation specific thresholds agreed better with FESM map.

[Figure]

Figure 7. Fire severity classification maps from FESM and predictions based on vegetation specific and fixed thresholds for wildfires in 2016 to 2019 in NSW.

Table 3 shows the confusion matrix for fire severity classification between FESM and predictions based on vegetation specific and fixed thresholds. It is noted that predictions based on vegetation specific thresholds exhibit better ability of classing extreme and high severity with accuracy of 0.64 and 0.76, respectively. While the classification accuracy for extreme and high severity of predictions based on fixed thresholds are 0.21 and 0.39, respectively. Predictions based on vegetation specific thresholds also have better accuracy of classifying moderate severity with value of 0.62, compared to those based on fixed thresholds with value of 0.47. Both predictions based on vegetation specific and fixed thresholds show poor performance in classifying low severity, with accuracy of 0.24 and 0.26 respectively. The overall classification accuracy for predictions based on vegetation specific thresholds is 0.57, which is better than predictions based on fixed specific thresholds with accuracy of 0.36.

Table 3. Confusion matrix for fire severity classification between FESM and predictions based on vegetation specific and fixed thresholds.

| | Vegetation specific | | | | | Fixed | | | |
| --- | --- | --- | --- | --- | --- | --- | --- | --- | --- |
| | Extreme | High | Moderate | Low | | Extreme | High | Moderate | Low |
| Extreme | 4345 | 2378 | 6 | 3 | Extreme | 1448 | 2822 | 2027 | 435 |
| High | 1490 | 6947 | 605 | 1 | High | 1430 | 3561 | 3358 | 694 |
| Moderate | 3 | 5702 | 9338 | 5 | Moderate | 998 | 4633 | 7084 | 2333 |
| Low | 0 | 172 | 7125 | 2372 | Low | 161 | 1722 | 5264 | 2522 |

Minor comments

Figure 1: it is a bit hard to visualize the 'wildfire for cross validation' in the map: is it underlaid
to the colored burned areas? I assume the burn years refer to the dataset mentioned in the
following page.

NSW National Parks and Wildlife Service 88 (NPWS) Fire History – Wildfire and Prescribed
Burns dataset (https://data.nsw.gov.au/data/dataset/1f694774-49d5-47b8- 89 8dd0-
77ca8376eb04 )

IF so, maybe mention this in the caption.

Also, it appears that the link is not working [I tried and accessed it on 05-feb-2024]

**Respond:** We have redesigned the Figure to make it clearer to see. We also mentioned the
source for the burn area map and fixed the link (https://datasets.seed.nsw.gov.au/dataset/fire-
history-wildfires-and-prescribed-burns-1e8b6 ).

[Figure]

Figure 1. Locations of study wildfires over New South Wales (NSW), Australia. The burn area is
from NSW National Parks and Wildlife Service (NPWS) Fire History – Wildfire and Prescribed
Burns dataset.

Paragraph from line 206-217: Figure 2 should be Figure 3, Same for the references in the
following chapters, it seems the authors refers to figure 3 as 2 (Eg line 221)

**Respond:** We have revised them accordingly.

Line 212: typo on the number, should be 6.7% not 6,7%

**Respond:** We have revised it accordingly.

Figure 3: are the vegetation numbers from n to 16 in figure b referring to the legend in figure a?
if so maybe leave only one legend to avoid confusion on what the number represents, or add the
names of vegetation on the x axis rather than as an additional color bar

**Respond:** We have redesigned the Figure 3.

[Figure]

(a)                                                                                    (b)

Figure 3. (a) The proportion of burnt area and (b) the distribution of fire severity grouped by vegetation type, over NSW from 2000 to 2019

References

Collins, L., Griffioen, P., Newell, G., Mellor, A., 2018. The utility of Random Forests for
wildfire severity mapping. Remote Sensing of Environment 216, 374–384.
https://doi.org/10.1016/j.rse.2018.07.005

aerial photos for validation. Maps produced using the RF classifier in GEE had similar spatial
patterns in fire severity classes as maps produced using time-consuming hand digitisation of
aerial images

Dixon, D.J., Callow, J.N., Duncan, J.M.A., Setterfield, S.A., Pauli, N., 2022. Regional-scale fire severity mapping of Eucalyptus forests with the Landsat archive. Remote Sensing of Environment 270, 112863. https://doi.org/10.1016/j.rse.2021.112863

aerial photo observations

Gale, M.G., Cary, G.J., van Dijk, A.I.J.M., Yebra, M., 2023. Untangling fuel, weather and management effects on fire severity: Insights from large-sample LiDAR remote sensing analysis of conditions preceding the 2019-20 Australian wildfires. Journal of Environmental Management 348, 119474. https://doi.org/10.1016/j.jenvman.2023.119474

Gibson, R., Danaher, T., Hehir, W., Collins, L., 2020. A remote sensing approach to mapping fire severity in south-eastern Australia using sentinel 2 and random forest. Remote Sensing of Environment 240, 111702. https://doi.org/10.1016/j.rse.2020.111702

Aerial photo interpretation classification of fire severity

Gorokhovich, Y., Voustianiouk, A., 2006. Accuracy assessment of the processed SRTM-based elevation data by CGIAR using field data from USA and Thailand and its relation to the terrain characteristics. Remote Sensing of Environment 104, 409–415. https://doi.org/10.1016/j.rse.2006.05.012

Hawker, L., Uhe, P., Paulo, L., Sosa, J., Savage, J., Sampson, C., Neal, J., 2022. A 30 m global map of elevation with forests and buildings removed. Environ. Res. Lett. 17, 024016. https://doi.org/10.1088/1748-9326/ac4d4f

Shortridge, A., Messina, J., 2011. Spatial structure and landscape associations of SRTM error. Remote Sensing of Environment 115, 1576–1587. https://doi.org/10.1016/j.rse.2011.02.017